# CodeDPO: Aligning Code Models with Self Generated and Verified Source Code

## Abstract

Code generation models have shown significant potential for programming tasks. However, existing training methods like supervised fine-tuning face key limitations: they do not effectively teach models to prioritize correct over incorrect solutions in ambiguous situations, nor do they effectively optimize the runtime efficiency of the generated code. To address these challenges, we propose CodeDPO, a framework that integrates preference learning into code generation to improve two key code preference factors: code correctness and efficiency. CodeDPO employs a novel dataset construction method, utilizing a self-generation-and-validation mechanism that simultaneously generates and evaluates code and test cases. The underlying assumption is that test cases executable by multiple code snippets provide more reliable validation, and code that passes more tests is more likely to be correct. Through this self-validation process, our PageRank-inspired algorithm iteratively updates the ranking score of each code snippet, ultimately creating a code preference optimization dataset based on correctness and efficiency. CodeDPO is flexible and scalable, generating diverse preference optimization data without depending on powerful models such as GPT-4. Through comprehensive evaluations of five widely used benchmarks, CodeDPO demonstrates significant improvements in correctness and efficiency compared to existing methods. Our experiments prove that CodeDPO enhances the capabilities of LLMs in code generation and provides a robust foundation for conducting code preference optimization in more complex and challenging real-world scenarios. [1]

## 1 Introduction

In recent years, code generation models have gained significant attention for their potential to automate software development. Models such as GPT-4 (GPT-4, 2023), Claude, and open-source alternatives like Phi (Gunasekar et al., 2023; Abdin et al., 2024), DeepSeekCoder (Guo et al., 2024), and StarCoder (Li et al., 2023; Lozhkov et al., 2024) have demonstrated the capability of LLMs to handle complex code generation tasks. However, one of the ongoing challenges lies in boosting the correctness and efficiency of the generated code.

To improve code generation models, a common approach is supervised fine-tuning (SFT) (Zhang et al., 2023b), where models are trained on pairs of instructions and correct code snippets. While SFT improves the overall quality of the generated code, it falls short in teaching models to consistently prefer correct solutions over incorrect ones (Hong et al., 2024). Figure 1 illustrates the likelihood of generating code with varying correctness and efficiency during SFT training. When we adopt SFT training on those correct solutions, as the likelihood of preferred outputs increases, the probability of generating undesirable outputs also rises, leading to performance saturation.

To address these limitations, recent research has turned to direct preference optimization (DPO) (Rafailov et al., 2024), a method designed for alignment based on pairwise preference data. DPO allows models to rank different outputs and choose preferable solutions (e.g., more factual or helpful). While DPO has shown success in reasoning tasks like mathematics (Lai et al., 2024; Wu et al., 2024), its application in code generation remains under-explored. Unlike natural language tasks, code generation requires objective metrics, such as executability, which poses challenges for directly applying

---

[1]Code and additional details are available: `https://anonymous.4open.science/r/CodeDPO/`

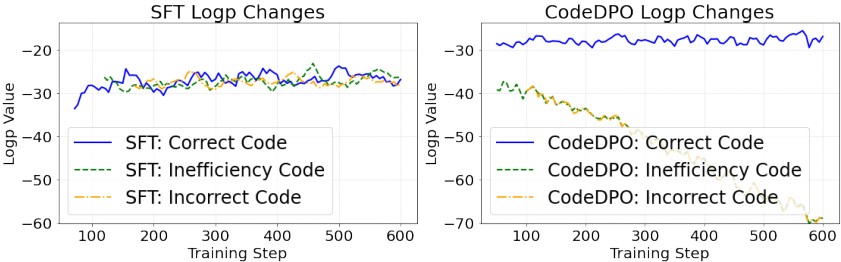

Figure 1: Log probabilities for code with varying correctness and efficiency during Phi-2-2.7B model training on our constructed dataset. The traditional SFT strategy struggles to teach models to prefer correct solutions over incorrect or slow ones. In contrast, our CodeDPO approach effectively optimizes for both correctness and efficiency.

DPO. **In this paper, we first define code preference based on two key factors—Correctness and efficiency.** Correctness refers to whether the code solves the problem accurately, while efficiency measures how quickly the code runs. Existing methods (Gee et al., 2024; Zhang et al., 2024) rely heavily on high-quality test cases to assess correctness. However, these approaches struggle to fully address correctness and efficiency, facing limitations such as restricted data diversity, an imbalance between positive and negative samples, and insufficient focus on optimizing code efficiency.

In this paper, we introduce CodeDPO, a novel framework that integrates preference learning into code model training to optimize both correctness and efficiency. CodeDPO constructs the dataset from real-world code repositories using a self-generation-and-validation mechanism, where code and test cases are simultaneously generated and evaluated. We assume that tests executable by more code snippets are more reliable, and code that passes more tests is more likely to be correct. To implement this, CodeDPO uses a mutual verification process: each receives an initial self-validation score, which is iteratively updated using a PageRank-inspired (Page, 1999) algorithm. This algorithm adjusts the credibility of each code snippet and tests by considering their relations in cross-verification, prioritizing solutions based on correctness and efficiency. The final preference-optimized dataset is then used to train various code models using the DPO learning algorithm. A key advantage of CodeDPO is its flexibility. Unlike existing methods that rely on high-quality test cases or powerful models to generate them, CodeDPO does not depend on these resources. Its self-generation and validation mechanism supports the scalable creation of diverse and robust preference optimization data. This allows our framework to optimize code models for real-world scenarios where high-quality test data may be sparse.

CodeDPO can serve as a crucial step in the post-training phase of code models. We conduct experiments on five popular benchmarks such as HumanEval (Chen et al., 2021), HumanEval+ (Liu et al., 2024a), MBPP (Austin et al., 2021), MBPP+, and DS-1000 (Lai et al., 2023) with CodeDPO, demonstrating its superiority over existing methods. Notably, we develop a top-performing 6.7B model by building on an existing SFT strategy (Guo et al., 2024; Wei et al., 2023) and further enhancing it with our CodeDPO approach, achieving an impressive 83.5% pass rate on HumanEval. We also conduct ablation studies to investigate the impact of our self-generation-and-validation mechanism and other preference optimization settings. Our findings confirm that CodeDPO enhances the code generation capabilities of LLMs while providing a solid foundation for further research into optimizing code generation for both correctness and efficiency.

## 2 RELATED WORK

### 2.1 LARGE LANGUAGE MODELS FOR CODE

Code generation, which automates writing source code from natural language (NL) descriptions, is gaining significant attention. LLMs have shown strong capabilities in this area due to their large-scale training on diverse datasets, such as OpenAI's GPT-4 (GPT-4, 2023), StarCoder (Li et al., 2023; Lozhkov et al., 2024), Code Llama (Rozière et al., 2023), and DeepSeekCoder (Guo et al., 2024). These models are often fine-tuned further, such as through instruction-supervised fine-tuning (SFT), to maximize their coding potential. Since gathering high-quality data is difficult, researchers

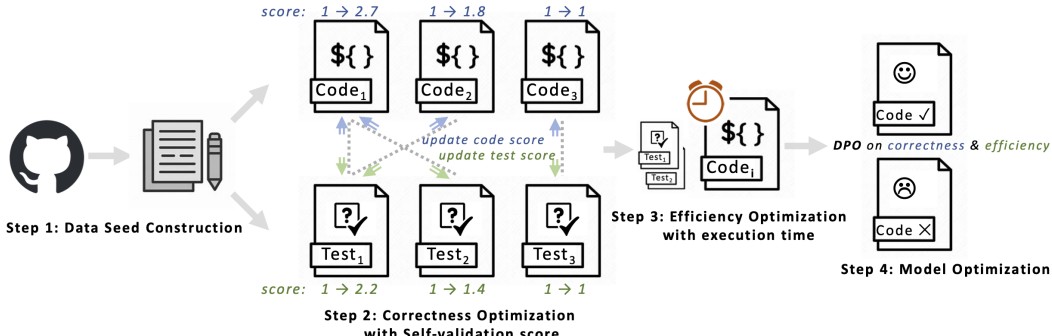

Figure 2: Our CodeDPO involves four steps: ❶ Data Seed Construction with real-world source code; ❷ Correctness Optimization with self-validation score (in this figure we set $T$ to 2 and $d$ to 0.5. For simplicity, the final score in the figure is rounded to one decimal place. Details are shown in Appendix H.3.1); ❸ Efficiency Optimization with execution time on credible tests; ❹ Model Optimization Training.

use self-instruct methods to generate synthetic instruction data from powerful models like GPT-4 (Wang et al., 2022; Taori et al., 2023; Chaudhary, 2023). Evol-Instruct (Luo et al., 2023) uses more complex prompts for better data generation. OSS-instruct (Wei et al., 2023) allows LLMs to get inspired from real-world code snippets for better quality in coding tasks. While these SFT methods boost code quality, it does not fully train models to prefer correct solutions over incorrect ones (Hong et al., 2024). Updating training strategies is critical for improving these code models to handle various coding tasks.

## 2.2 PREFERENCE OPTIMIZATION FOR CODE MODELS

Preference optimization techniques have recently been used to help LLMs prefer better outputs over weaker ones in various natural language tasks (Rafailov et al., 2024). The **Direct Preference Optimization (Rafailov et al., 2024)** has been widely applied to LLM alignment due to its convenience and effectiveness. Its objective is defined as:

$$L_{\text{DPO}} = -\mathbb{E}_{(x,y_w,y_l)\sim\mathcal{D}} \left[ \log \sigma \left( \beta \log \frac{\pi_\theta(y_w \mid x)}{\pi_{\text{ref}}(y_w \mid x)} - \beta \log \frac{\pi_\theta(y_l \mid x)}{\pi_{\text{ref}}(y_l \mid x)} \right) \right]$$

Compared with the SFT loss, the DPO loss introduces a preference-based mechanism. Instead of merely maximizing the likelihood of ground truth data, as in SFT, DPO optimizes the model to align with human preferences by leveraging both preferred responses ($y_w$, winning) and dispreferred responses ($y_l$, losing). While DPO has proven effective in reasoning tasks like mathematics (Lai et al., 2024), its use in code generation is still under-explored. Code generation requires objective measures of correctness and efficiency, unlike natural language tasks where preferences are often more subjective. Some works have simply explored PO. Code-Optimize (Gee et al., 2024) builds its dataset from the MBPP-train subset, which includes just 384 problems. PLUM uses GPT-4 to generate tests, which are then used to validate and rank code solutions. PLUM currently achieves state-of-the-art performance in preference optimization for code models. However, PLUM (Zhang et al., 2024) faces some limitations. It uses a limited number of tests to validate the code, and the resulting dataset is imbalanced due to its validation method, which means it can only use KTO (Ethayarajh et al., 2024) for training. Additionally, PLUM does not consider the code efficiency. This paper introduces CodeDPO, which does not rely on external test cases or powerful models for dataset generation. Our approach uses a self-generation and validation mechanism to create balanced preference pairs, aiming to optimize both correctness and efficiency.

## 3 CODEDPO: SELF-VERIFIED PERFORMANCE OPTIMIZATION CODE GENERATION FRAMEWORK

CodeDPO is designed to integrate preference learning into code generation models, improving both the correctness and efficiency of the generated code. As shown in Figure 2, our method involves

four key steps: ❶ **Data Seed Construction with real-world source code**: We first collect a data seed from open-source code repositories and generate programming task prompts. ❷ **Correctness Optimization with self-validation score**: We generate code and tests simultaneously, using a self-generation-and-validation loop to build a dataset for correctness optimization. The self-validation score is iteratively updated based on whether the generated code passes the tests. We assume that tests executable by multiple code snippets are more reliable, and code that passes more tests is more likely to be correct. As illustrated in the figure, after two iterations, the score of *code-1* changes from *1* to *1.75* to *2.6875* ($\sim$*2.7* in the figure), as it passes more reliable tests and receives higher scores with each update, indicating a greater likelihood of correctness. ❸ **Efficiency Optimization with execution time**: We measure execution time on selected credible test sets to build the dataset for efficiency optimization. In the figure, we select *test-1* and *test-2* as the credible test set to measure the execution time of each code snippet. ❹ **Model Optimization Training**: We collect the dataset from the previous two stages and use Direct Preference Optimization (DPO) to train various code models.

## 3.1 DATA SEED CONSTRUCTION

The data seed construction for CodeDPO is the first step for initiating the preference learning process to generate programming task prompts. We adopt a method inspired by OSS-instruct (Wei et al., 2023; 2024)[2], which extracts key programming concepts from open-source code repositories. These concepts serve as the foundation for generating various programming task prompts. For example, a code snippet that performs sorting operations might highlight concepts such as sorting algorithms, data structure traversal, and time complexity. From these concepts, we generate code generation prompts. The data seed thus allows the model to explore a wide range of scenarios.

## 3.2 CORRECTNESS OPTIMIZATION WITH SELF-GENERATION AND VALIDATION

Central to CodeDPO is the self-generation-and-validation loop, which enables the model to iteratively update the code correctness rank through mutual validation of code and test cases (Chen et al., 2022; 2023; Zhang et al., 2023a). The process begins by generating multiple candidate code snippets based on a prompt. Simultaneously, corresponding test cases are generated to evaluate these snippets. The validation loop follows these steps: **1. Code Generation:** Given an instruction, the model generates a set of candidate code snippets $C = \{c_1, c_2, ..., c_n\}$. **2. Test Case Generation:** Test cases $T = \{t_1, t_2, ..., t_m\}$ are generated in parallel to validate the candidate snippets. **3. Validation Process:** Each code snippet is executed against the generated test cases. The validation outcomes are used to update the self-validation scores for both the code snippets and the test cases.

**Ranking Code Snippets and Test Cases Using Self-Validation Scores** To rank both code snippets and tests, we employ a PageRank-inspired (Page, 1999) iterative algorithm. Initially, each code and test is assigned a self-validation score of 1. Over a fixed number of iterations $T = 10$, these scores are updated based on the performance of the snippets and test cases during validation.

The self-validation score for code snippets and test cases is updated using the following formulas:

$$\text{Score}_t(c_i) = (1 - d) \times \text{Score}_{t-1}(c_i) + d \times \sum_{t_j} \text{Score}_{t-1}(t_j) \times \text{Link}(t_j, c_i) \tag{1}$$

$$\text{Score}_t(t_j) = (1 - d) \times \text{Score}_{t-1}(t_j) + d \times \sum_{c_i} \text{Score}_{t-1}(c_i) \times \text{Link}(c_i, t_j) \tag{2}$$

Where $d$ is the damping factor, and $\text{Link}(t_j, c_i)$ indicates whether a code snippet $c_i$ passes the test case $t_j$. This iterative process is repeated until convergence. After $T$ iterations, the final rankings reflect the quality of the code snippets and test cases based on the correctness.

---

[2]We follow the implementation provided at `https://github.com/bigcode-project/starcoder2-self-align/tree/fd0af77e2773b14696c7cea02a472f9e99d9c4e3`.

### 3.3 Execution Efficiency Optimization

In addition to ensuring correctness, CodeDPO integrates execution efficiency optimization to ensure that our approach generates functionally correct and efficient code. During the self-validation loop, the execution time for each code snippet is recorded. However, not all test cases accurately reflect the efficiency of the code. To address this, we use the top-performing code from the correctness optimization phase as a reference, assuming the test cases it passes are credible. The total execution time for each code snippet is then measured based on the subset of these credible tests. Code snippets that pass these credible test cases with lower execution times are assigned higher efficiency scores. Finally, we collect both fast and slow code snippets as part of the training dataset for execution efficiency optimization, which is used for further training, encouraging the model to prioritize solutions that are accurate and optimized for speed during code generation.

### 3.4 Final Dataset and Model Optimization

The final dataset is built from the previous two optimization dataset construction stages, accounting for correctness and execution time. This dual-optimization approach ensures that our CodeDPO dataset can train models to generate not only accurate code but also efficient solutions, addressing both functional and performance challenges in real-world coding tasks. We filter out samples whose ranking scores are identical or too close. The final dataset consists of **93k correctness optimization samples** and **21k efficiency optimization samples**. Each sample includes a unique code problem prompt with a preferred and a rejected solution. We carefully avoid overlap between the data seeds of correctness and efficiency samples, ensuring that the constructed dataset captures various problems and instructions. In the subsequent training, we combine both correctness and efficiency data to optimize the model in both aspects simultaneously.

In our experiments, we apply Direct Preference Optimization (DPO) (Rafailov et al., 2024) across various code models to facilitate optimization learning. To enhance the stability and robustness of the training process, we employ RPO (Pang et al., 2024; Liu et al., 2024b) format loss, which essentially consists of a weighted SFT loss on the chosen preferences together with the original DPO loss, which is defined as: $L = L_{DPO} + L_{SFT}$. CodeDPO is plug-and-play and can be applied to nearly all code models, regardless of their type or training stage. We utilize both base models and SFT models as the backbone for further training. Our goal is to demonstrate that CodeDPO has the potential to enhance code models at different stages of their training, even for models that have undergone extensive training or fine-tuning. The setup details are provided in Section 4.2.

## 4 Experiment Setup

In this study, we aim to investigate the following research questions:

**RQ1: Does CodeDPO improve the correctness of generated code compared to baseline models on standard benchmarks? How does CodeDPO compare with other code preference optimization baselines?** We evaluate the pass rate of CodeDPO on benchmarks such as HumanEval (Chen et al., 2021), HumanEval+ (Liu et al., 2024a), MBPP (Austin et al., 2021), MBPP+, and DS-1000 (Lai et al., 2023). We further compare the performance of CodeDPO with other baselines (Gee et al., 2024; Zhang et al., 2024) that also utilize preference optimization techniques.

**RQ2: Does CodeDPO enhance the execution efficiency of generated code?** We measure the execution efficiency of code generated by CodeDPO compared to baseline models.

**RQ3: What is the impact of the self-generation-and-validation algorithm on CodeDPO's performance?** We perform ablation studies by removing or modifying the self-generation-and-validation mechanism to assess its contribution to the overall performance.

**RQ4: How does the choice of preference optimization strategy affect CodeDPO's effectiveness?** We evaluate different preference optimization strategies, including Direct Preference Optimization (DPO), Kahneman-Tversky Optimization (KTO) (Ethayarajh et al., 2024), and Supervised Fine-Tuning (SFT), to understand their impact on the model's performance.

**RQ5: How does data scaling influence the performance of CodeDPO?** We investigate data scaling for CodeDPO by varying the amount of training data to show how data size affects its ability.

## 4.1 BACKBONE LLMS

We evaluate several widely used LLMs in the code generation domain for our experiments, covering both **base models** and **SFT models** at different training stages. For **base models**, we apply CodeDPO to **Phi-2 (2.7B)** (Gunasekar et al., 2023), **DeepSeekCoder-base (1.3B, 6.7B)** (Guo et al., 2024), and **StarCoder2-base (7B)** (Lozhkov et al., 2024). Additionally, we evaluate our method on several fine-tuned **SFT models** (Wei et al., 2023), including **Magicoder-CL-7B**, **Magicoder-S-CL-7B**, **Magicoder-DS-6.7B**, and **Magicoder-S-DS-6.7B**, which are fine-tuned based on *CodeLlama-7B* and *DeepSeekCoder-base-6.7B* using state-of-the-art SFT techniques.

While applying the PO phase after SFT is generally recommended (Rafailov et al., 2024), we extend our evaluation to base models as they can generate more diverse code snippets and offer more significant potential for improvement (Wang et al., 2024). Since CodeDPO's optimization focuses on objective metrics such as code correctness and efficiency, it contrasts with other natural language tasks where preferences are often more subjective. This does not require our backbone model to have a strong ability to follow subjective instructions, allowing CodeDPO to be directly applied to base models. We choose all these popular models as the backbone of our experiments to optimize correctness and execution efficiency.

## 4.2 TRAINING AND INFERENCE SETTINGS

For dataset construction, in order to balance generation speed and cost efficiency, we use *DeepSeekCoder-v2* as the data generation model. For each problem prompt, we sample 15 code solutions and test cases from this model with $temperature = 1.5$. To construct the preference optimization dataset, we set $T$ to 10 and $d$ to 0.85 for the self-validation score. Our practice shows that this parameter configuration quickly yields a stable ranking score. In this paper, we focus on constructing a Python dataset. The total cost of our dataset construction process is nearly 80$.

For training, we train each code model for 10 epochs and select the best-performing model based on the lowest validation loss. We utilize a learning rate of 5e-6 with a linear scheduler and warm-up. The maximum sequence length is set to 2048 tokens.

For inference, we use greedy search decoding for code generation. All evaluations use the framework from *bigcode-evaluation-harness* (Ben Allal et al., 2022). We use 16 A100 GPUs for all experiments.

## 5 RESULTS AND ANALYSES

### 5.1 CODE CORRECTNESS (RQ1)

To answer **RQ1**, we evaluate the model performance on five widely-used code generation benchmarks: **HumanEval**, **HumanEval+**, **MBPP**, **MBPP+**, and **DS-1000**. Following the standard training process (base model → SFT → DPO), we first record the performance of the base model, SFT model, and DPO-aligned model on DeepSeekCoder-6.7B, as shown in Table 1. With the enhancement of our CodeDPO, the final model achieves an 83.5% pass rate on HumanEval. Notably, even after high-quality SFT training, CodeDPO still achieves additional performance improvements. CodeDPO plays a crucial role in the post-training phase of code models, significantly boosting overall performance.

| Model | HumanEval | HumanEval+ | MBPP | MBPP+ |
|---|---|---|---|---|
| DeepSeekCoder-6.7B-base | 47.60 | 39.60 | 70.20 | 56.60 |
| + SFT *(with MagiCoder-OSS-instruct)* | 73.17 | 68.29 | 76.72 | 66.67 |
| + SFT + **Our CodeDPO** | **83.54** | **76.22** | **80.70** | **70.93** |

Table 1: Pass rates (%) of code models at different stages on HumanEval(+) and MBPP(+). We track the performance of the base model, SFT model, and DPO-aligned model on DeepSeekCoder-6.7B. Our CodeDPO shows additional improvements, even after high-quality SFT training.

We further evaluate the performance of CodeDPO alongside baselines[3] on a wide range of models, including four base models and four SFT models. As shown in Table 2, CodeDPO achieves the best performance on both HumanEval(+) and MBPP(+). Compared to the baseline models in the first row of each block, we observe that CodeDPO delivers significant improvements across all models, regardless of their initial performance. Notably, we achieve a 36.1% relative improvement on StarCoder2-7B. Additionally, CodeDPO shows remarkable gains on the more challenging HumanEval+, demonstrating its robustness under stricter evaluation. Thanks to CodeDPO's data construction strategy, we can build a reliable preference dataset that helps the model favour high-quality outputs, leading to more robust and reliable code generation.

For DS-1000, as shown in Table 3, we further evaluate CodeDPO across different libraries. We did not incorporate prior knowledge of specific Python libraries in our data construction. Thanks to our approach's flexibility, we can create a wide variety of programming problems and corresponding code pairs. While we observe slight performance drops in the Torch and TensorFlow settings, this may be due to the relatively low percentage of these libraries in our dataset construction. However, CodeDPO demonstrates overall performance improvements over their respective baselines. It is important to note that DS-1000 differs from benchmarks like HumanEval and MBPP in data format and the coding skills it assesses. The dataset generation process for DS-1000 ensures that *it is excluded from nearly all models' training sets*, making the improvements we observe on DS-1000 reliable. These results show that CodeDPO does more than just adapt to standard coding benchmarks like HumanEval. It proves that CodeDPO can enhance the model's coding capabilities in more complex and diverse scenarios.

## 5.2 CODE EFFICIENCY (RQ2)

To address **RQ2**, we follow existing methods (Shypula et al., 2024) by measuring the execution time of the generated code and calculating the **speed-up** ratio. We also evaluate the **percentage of optimized code** before and after applying CodeDPO, where a program is considered optimized if it is at least 10% faster than its baseline. These metrics are based on the intersection of solved problems before and after applying CodeDPO. We select **HumanEval+** and **MBPP+** for evaluation because they significantly expand the diversity of test cases, making them more reliable for measuring the execution efficiency of the generated code under a variety of edge cases. Since runtime environments can affect measurements, we repeat each evaluation five times and show the distribution in Figure 3. It is clear that CodeDPO consistently improves code performance. The speed-up ratio shows that our method speeds up the code by 1.25 to 1.45 times. The range in the figure highlights that most measurements cluster around a significant performance boost. Additionally, the percentage of optimized code indicates that after applying CodeDPO, around 20%-45% of generated code solutions have been improved, confirming its effectiveness in enhancing code efficiency.

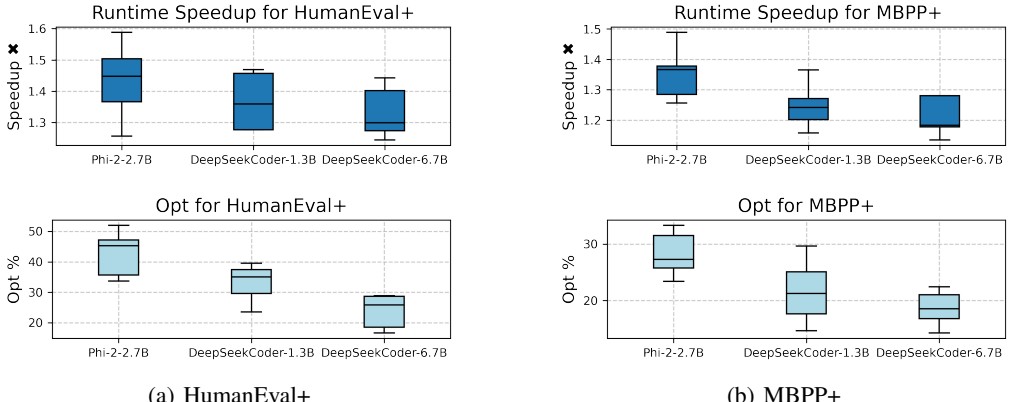

(a) HumanEval+         (b) MBPP+

Figure 3: Runtime Speedup and Percentage of Optimized Code on HumanEval+ and MBPP+.

[3]The baselines have not yet published their datasets. We reproduced the Code-Optimize experiment based on the reported settings. For PLUM, we report results from their paper using models identical to ours, which is why some models do not include PLUM results.

| Model | HumanEval | HumanEval+ | MBPP | MBPP+ |
|---|---|---|---|---|
| ***SFT Model*** | | | | |
| MagiCoder-CL-7B | 51.21 | 48.78 | 65.60 | 55.82 |
| Our CodeDPO | **60.36** | **54.87** | **70.93** | **59.15** |
| *Code-Optimise* | 48.78 | 46.95 | 67.17 | 57.14 |
| MagiCoder-S-CL-7B | 67.07 | 61.59 | 69.58 | 60.58 |
| Our CodeDPO | **74.39** | **71.95** | **71.43** | **61.40** |
| *Code-Optimise* | 64.63 | 54.88 | 69.42 | 60.15 |
| *PLUM* | 73.80 | 69.50 | 71.40 | 60.80 |
| MagiCoder-DS-6.7B | 57.93 | 53.66 | 75.93 | 64.02 |
| Our CodeDPO | 67.07 | 62.80 | **81.70** | **68.92** |
| *Code-Optimise* | 57.93 | 51.83 | 76.19 | 64.91 |
| *PLUM* | **71.30** | **65.90** | 79.60 | 66.70 |
| MagiCoder-S-DS-6.7B | 73.17 | 68.29 | 76.72 | 66.67 |
| Our CodeDPO | **83.54** | **76.22** | **80.70** | **70.93** |
| *Code-Optimise* | 68.90 | 64.63 | 78.20 | 67.92 |
| *PLUM* | 80.50 | 73.80 | 80.40 | 69.30 |
| ***Base Model*** | | | | |
| Phi-2-2.7B | 48.78 | 46.34 | 65.34 | 54.49 |
| Our CodeDPO | **57.32** | **51.83** | **69.05** | **56.88** |
| *Code-Optimise* | 49.39 | 47.56 | 67.42 | 55.80 |
| DeepSeekCoder-1.3B | 31.53 | 28.65 | 57.40 | 48.67 |
| Our CodeDPO | **42.07** | **38.04** | **61.37** | **53.43** |
| *Code-Optimise* | 34.15 | 30.49 | 59.15 | 49.87 |
| DeepSeekCoder-6.7B | 47.60 | 39.60 | 70.20 | 56.60 |
| Our CodeDPO | **59.75** | **51.83** | 72.18 | **60.01** |
| *Code-Optimise* | 47.56 | 37.20 | 72.18 | 57.64 |
| *PLUM* | 56.70 | 48.80 | **72.90** | 58.90 |
| StarCoder2-7B | 35.40 | 29.90 | 54.40 | 45.60 |
| Our CodeDPO | **48.17** | 34.15 | 58.40 | **49.37** |
| *Code-Optimise* | 32.32 | 28.05 | 58.90 | 47.89 |
| *PLUM* | 46.30 | **39.60** | **60.40** | 49.10 |

Table 2: Pass rate (%) of CodeDPO compared to baseline models on HumanEval and MBPP.

## 5.3 ABLATION STUDIES

### 5.3.1 SELF GENERATION AND VALIDATION ALGORITHM (RQ3)

**Correlation between self-validation scores and actual code accuracy using HumanEval ground truth tests** To evaluate the effectiveness of our self-generation-and-validation algorithm, we examine the correlation between self-validation scores and actual code accuracy. We use a benchmark with pre-existing ground truth test cases, such as HumanEval, for *this preliminary experiment*. For each problem in HumanEval, we sample 15 code solutions and tests following the setting in Section 4, and then use different strategies to rank these generated codes. To evaluate the rank quality, we execute with the ground truth for each code to get the actual code accuracy. Then, we calculate the correlation between our self-validation scores and actual code accuracy.

We consider three experimental strategies: ❶ **Self-validation score**, which refers to our original method. ❷ **Filter with all tests**, which assumes all generated test cases are correct and uses them to judge code correctness. This approach creates passed/non-passed pairs, similar to the baseline **PLUM** (though PLUM uses GPT-4 for test generation, while we use a more cost-effective model). ❸ **Sort by number of passed tests**, which counts the number of passed tests for each code among all generated tests, using the code with the most and least passed tests as the comparison pair. This principle is commonly employed in post-processing methods, such as **CodeT** Chen et al. (2022).

| Model | plot (155) | np (220) | pd (291) | torch (68) | scipy (106) | sk (115) | tf (45) | Average |
|---|---|---|---|---|---|---|---|---|
| *SFT Model* | | | | | | | | |
| Magic-CL-7B | 54.8 | 16.4 | 16.5 | 17.6 | 23.6 | 29.6 | **33.3** | 25.5 |
| Our CodeDPO | **57.4** | **37.3** | **22.7** | **22.1** | **35.8** | **31.3** | 31.1 | **34.0** |
| Magic-S-CL-7B | 52.3 | 43.2 | 30.6 | **47.1** | 34.9 | **46.1** | **44.4** | 40.7 |
| Our CodeDPO | **58.7** | **44.5** | **31.3** | 38.2 | **40.6** | 42.6 | 33.3 | **41.3** |
| Magic-DS-6.7B | 55.5 | 37.7 | 28.2 | **25.0** | 34.0 | **45.2** | **33.3** | 37.1 |
| Our CodeDPO | **59.4** | **40.5** | **29.2** | 23.5 | **39.6** | 42.6 | 31.1 | **38.7** |
| Magic-S-DS-6.7B | 53.5 | 49.5 | 30.6 | **47.1** | 35.8 | **53.0** | **40.0** | 42.9 |
| Our CodeDPO | **59.4** | **50.5** | **31.9** | 39.7 | **41.5** | 47.8 | 33.3 | **43.7** |
| *Base Model* | | | | | | | | |
| Phi-2-2.7B | 42.6 | 33.6 | 15.5 | **16.2** | 17.0 | 11.3 | **17.8** | 23.5 |
| Our CodeDPO | **49.0** | 33.6 | **16.5** | 14.7 | **20.8** | **14.8** | 13.3 | **25.3** |
| DSC-1.3B | **36.8** | 19.5 | 10.0 | 14.7 | 10.4 | **17.4** | **11.1** | 17.5 |
| Our CodeDPO | 34.8 | **23.6** | **10.7** | **14.7** | **20.8** | 13.9 | 8.9 | **18.9** |
| DSC-6.7B | 52.3 | 35.5 | 20.6 | **19.1** | 24.5 | **37.4** | **22.2** | 31.1 |
| Our CodeDPO | **56.8** | **36.4** | **21.6** | 17.6 | **34.0** | 34.8 | 20.0 | **32.8** |
| StarCoder2-7B | 54.2 | 37.7 | 18.6 | **25.0** | 31.1 | 23.5 | **35.6** | 31.4 |
| Our CodeDPO | **56.8** | **38.2** | **18.9** | 20.6 | **39.6** | **25.2** | 31.1 | **32.6** |

Table 3: Pass rate (%) of CodeDPO on DS-1000 across seven libraries using greedy decoding.

Table 4 presents the Spearman, Kendall's Tau, and Normalized Discounted Cumulative Gain (NDCG) metrics for the different ranking strategies. Our experiments show that the self-validation score is highly correlated with actual code accuracy, and its ranking closely reflects true code quality, making it a reliable metric for preference optimization. In contrast, filtering by all tests heavily depends on the quality of the test generation model. While baselines like PLUM ensure high-quality test generation using GPT-4, our more economical approach highlights that using all tests indiscriminately can introduce noise, as lower-quality tests skew the final ranking and poison the dataset. Sorting by the number of passed tests treats all tests equally important. However, due to the inherent uncertainty in generated tests, this method can be vulnerable to low-quality tests. Our proposed self-validation method employs a mutual reinforcement mechanism to update the credibility of both code and tests, effectively mitigating these issues.

| Method | Spearman | Kendall's Tau | NDCG |
|---|---|---|---|
| Self-validation score | **0.8598** | **0.8047** | **0.9653** |
| Filter with all tests | 0.6114 | 0.6114 | 0.8753 |
| Sort by # of passed tests | 0.7724 | 0.7250 | 0.9162 |

Table 4: Correlation between self-validation score and actual code accuracy on HumanEval.

**Impact of self-validation score on model performance** We apply these strategies to construct datasets and evaluate the final model performance in code generation. Table 5 presents the model performance across various dataset construction strategies. In addition, we introduce a new strategy—*random selection*—which randomly selects two code solutions from the generated code as the preference pair. The experiment results demonstrate that the self-generation-and-validation algorithm plays an essential role in ensuring the correctness and reliability of the preference dataset construction, significantly improving the performance of our CodeDPO framework.

### 5.3.2 IMPACT OF PO TRAINING STRATEGY (RQ4)

We explore the impact of different preference optimization strategies (DPO, KTO, and SFT) on model performance. For training, the SFT strategy uses the best code solution from our constructed

| Model | HumanEval | HumanEval+ | MBPP | MBPP+ |
|---|---|---|---|---|
| Phi-2-2.7B | 48.78 | 46.34 | 65.34 | 54.49 |
| Our CodeDPO | **57.32** | **51.83** | **69.05** | **56.88** |
| Filter with all tests | 49.39 | 48.17 | 69.17 | 55.13 |
| Sort by # of passed tests | 50.60 | 49.39 | 67.16 | 54.88 |
| Random selection | 22.56 | 18.90 | 45.11 | 36.59 |
| DeepSeekCoder-1.3B | 31.53 | 28.65 | 57.40 | 48.60 |
| Our CodeDPO | **42.07** | **38.04** | **61.37** | **53.43** |
| Filter with all tests | 34.75 | 29.89 | 57.40 | 48.80 |
| Sort by # of passed tests | 37.19 | 31.09 | 58.39 | 50.37 |
| Random selection | 21.34 | 18.29 | 48.94 | 38.35 |

Table 5: Ablations of our self validation score on the trained model performance.

dataset. In our KTO strategy, we replace DPO with KTO in our framework. As shown in Figure 1, the traditional SFT strategy struggles to guide the model in preferring correct solutions over incorrect or slower ones during training. The results in Table 6 demonstrate that DPO performs best among these strategies. Benefiting from our dataset construction method, we can obtain well-balanced preference pairs, enhancing the contrastive mechanism in DPO.

| Model | HumanEval | HumanEval+ | MBPP | MBPP+ |
|---|---|---|---|---|
| Phi-2-2.7B | 48.78 | 46.34 | 65.34 | 54.49 |
| SFT | 55.49 | 49.22 | 66.87 | 55.76 |
| Our CodeDPO | **57.32** | **51.83** | **69.05** | **56.88** |
| Our CodeKTO | 54.88 | 51.22 | 64.91 | 53.63 |
| DeepSeekCoder-1.3B-base | 31.53 | 28.65 | 57.40 | 48.67 |
| SFT | 39.02 | 35.36 | 59.45 | 50.26 |
| Our CodeDPO | **42.07** | **38.04** | **61.37** | **53.43** |
| Our CodeKTO | 40.85 | 35.98 | 59.65 | 50.13 |
| DeepSeekCoder-6.7B-base | 47.60 | 39.60 | 70.20 | 56.60 |
| SFT | 56.09 | 46.95 | 70.18 | 56.88 |
| Our CodeDPO | **59.75** | **51.83** | **72.18** | **60.01** |
| Our CodeKTO | 54.88 | 49.39 | 71.93 | 58.65 |

Table 6: Comparison of preference optimization strategies (DPO vs. KTO vs. SFT).

## 5.4    DATA SCALING LAW FOR CODEDPO (RQ5)

To address **RQ5**, we explore how scaling the training data affects CodeDPO's performance. As shown in Table 7, increasing the data consistently improves model performance, but these improvements gradually plateau as the dataset size grows. In our experiments, we carefully balance performance gains and training costs, ensuring optimal results with CodeDPO. Details are shown in Appendix A.

## 6    CONCLUSION

We propose CodeDPO, a preference optimization framework for code models that focuses on both correctness and efficiency. CodeDPO introduces a novel dataset construction method that utilizes a self-generation-and-validation mechanism, enabling the simultaneous generation and evaluation of code and test cases to ensure correctness. Our PageRank-inspired algorithm iteratively updates the self-validation score of each code snippet, prioritizing solutions based on correctness and efficiency. Our work technically validates the reliability of self-validation to synthesize preference optimization data, eliminating the need for complex resources such as pre-existing tests or powerful generation models. We hope this work opens new avenues for synthesizing data and implementing large-scale preference optimization for code models.

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

# A    DATA SCALING LAW FOR CODEDPO (RQ5)

We show the experiment results for RQ5, which can help us explore how scaling the training data affects CodeDPO's performance. We train the model with varying amounts of data—25%, 50%, and 75%—and evaluate its impact on the model performance. For example, HumanEval scores rise from 32.92 (25%) to 41.46 (75%), with similar trends observed on MBPP. In our experiments, we carefully balance performance gains and training costs, ensuring optimal results with CodeDPO. In further research, we plan to expand the current training scale to explore the extreme limits of CodeDPO's performance.

| Model | HumanEval | HumanEval+ | MBPP | MBPP+ |
|---|---|---|---|---|
| DeepSeekCoder-1.3B-base | 31.53 | 28.65 | 57.40 | 48.67 |
| 25% | 32.92 | 29.87 | 55.13 | 47.87 |
| 50% | 36.59 | 31.70 | 58.14 | 49.87 |
| 75% | 41.46 | 37.80 | 60.65 | 52.63 |
| Our CodeDPO | 42.07 | 38.04 | 61.37 | 53.43 |

Table 7: Model Performances with different data scaling in our CodeDPO.

# B    CODEDPO DATASET CONSTRUCTION ALGORITHM DESCRIPTION

In order to make it clear, we give a formal algorithm description of the CodeDPO construction pipeline in Algorithm 1.

# C    LLM PROMPTS FOR CODEDPO DATASET CONSTRUCTION

We use the following prompts for dataset seed construction and self-validation. During dataset construction, we first use code snippets from a randomly selected subset of **the Stack v1** dataset as input and prompt the LLM to generate the concept (LLM Prompt 1). Based on the concept, we then prompt the LLM to generate the task description (LLM Prompt 2).

For the validation process, we directly prompt the LLM with the task description to generate code solutions. Additionally, we prompt the LLM to generate only assertion statements as test cases (LLM Prompt 3). Since our chosen generation LLM is efficient and cost-effective, the entire process of data generation and construction takes around 40 hours on a server with 32 CPUs.

---

**LLM Prompt 1 for Concept Generation**

Extract key programming concepts from a given code snippet collected from the open source repositories. Present the concepts as a comma separated list.

`{Few-shot Examples}`

## Example 2
### Snippet

`{Input Code}`

### Concepts
`{need to generate}`

---

---

**Algorithm 1** CodeDPO Dataset Construction Pipeline

---

1: **procedure** CODEDPO(model, instruction, max_iterations)
2:     **Seed Construction:**
3:     Extract key programming concepts from source code repositories
4:     Generate code generation prompts and corresponding test cases
5:     Generate initial dataset $(instruction, solutions, testcases)$
6:     **Initialization:**
7:     Generate initial code snippets $C = \{c_1, c_2, ..., c_n\}$ from the instruction
8:     Generate test cases $T = \{t_1, t_2, ..., t_m\}$ corresponding to the instruction
9:     Initialize self-validation scores for code snippets and test cases: $\text{Score}(c_i) \leftarrow 1$, $\text{Score}(t_j) \leftarrow 1$
10:    Set damping factor $d \leftarrow 0.85$
11:    $i \leftarrow 0$
12:    **Self-Validation Loop:**
13:    **while** $i < \text{max\_iterations}$ **do**
14:       **for** each $c_i \in C$ **do**
15:         Execute $c_i$ on test cases $T$
16:         **for** each $t_j \in T$ **do**
17:           **if** $c_i$ passes $t_j$ **then**
18:             Update $\text{Score}(c_i)$ using Equation (1)
19:             Update $\text{Score}(t_j)$ using Equation (2)
20:             **Execution Time Optimization:**
21:             Record execution time for $c_i$
22:           **if** $c_i$ fails $t_j$ **then**
23:             Set execution time to max penalty to penalize $c_i$
24:           **end if**
25:         **end if**
26:         **end for**
27:       **end for**
28:       $i \leftarrow i + 1$
29:    **end while**
30:    **Final Dataset Collection:**
31:    **Correctness Optimization:**
32:    Select top-ranked code $c_{\text{top}}$ and low-ranked code $c_{\text{low}}$ for each instruction
33:    Store as dataset entries $(instruction, c_{\text{top}}, c_{\text{low}})$
34:    **Execution Time Optimization:**
35:    Select fastest code $c_{\text{fast}}$ and slowest code $c_{\text{slow}}$ for each instruction
36:    Store as dataset entries $(instruction, c_{\text{fast}}, c_{\text{slow}})$
37:    **return** final dataset entries
38: **end procedure**

---

> **LLM Prompt 2** for Task Prompt Generation
>
> Create a set of independent code instructions that are original, different, diverse, and high-quality, where the properties control an instruction's category, language, concepts, and difficulty.
>
> {Few-shot Examples}
>
> ## Example 2
> ### Property
>
> {Input Concept generated from the previous step}
>
> ### Instruction
> {need to generate}

---

**LLM Prompt 3 for Test Case Generation**

Generate only assertion statements based on the following description. Do not generate any other code:

`{Instruction}`

Generated Assertions:
`assert {need to generate}`

---

## D  EXPERIMENTS ON CHALLENGING CODE GENERATION TASKS

We conducted additional experiments on **LiveCodeBench** (Jain et al., 2024), one of the most challenging benchmarks for competitive coding tasks. The results, summarized below, will be included in the revised paper along with more model comparisons:

| Model | Easy | Medium | Hard |
|---|---|---|---|
| **Base Model** | | | |
| DeepSeek-Coder-6.7B | 39.9 | 7.4 | 0.4 |
| Our CodeDPO | 51.9 | 12.2 | 0.7 |
| **SFT Model** | | | |
| MagiCoder-S-DS-6.7B | 48.1 | 10.7 | 0.1 |
| Our CodeDPO | 53.1 | 16.3 | 0.7 |

Table 8: Performance comparison on LiveCodeBench across difficulty levels.

The results indicate that CodeDPO demonstrates significant performance improvements for both the base model and the supervised fine-tuning (SFT) model across all difficulty levels. The gains are particularly notable in the "medium" and "hard" subsets, which represent some of the most challenging problems in competitive programming tasks. These subsets often require a deep understanding of problem requirements and the ability to generalize to unseen scenarios.

These findings underscore the robustness and generalizability of CodeDPO, even in restricted evaluation settings such as LiveCodeBench. This highlights the effectiveness of the proposed framework for real-world, complex coding tasks.

## E  EXPERIMENTS ON CHALLENGING CODE EFFICIENCY TASKS

To evaluate code efficiency comprehensively, additional experiments are conducted on **EffiBench** (Huang et al., 2024). Since the absolute values of the results may vary depending on the specific execution environment, the analysis focuses on the relative improvements achieved by CodeDPO. The results are summarized in the table below and will be included in the revised paper alongside evaluations on additional models.

The results indicate that CodeDPO significantly reduces execution time and memory usage, both in absolute terms and after normalization, while maintaining comparable maximum memory usage. These improvements highlight the effectiveness of CodeDPO in optimizing code for both computational efficiency and resource usage, ensuring applicability to environments where performance and memory constraints are critical.

## F  EXECUTION TIME FOR CODE EFFICIENCY EXPERIMENTS

We present the average execution time (in seconds) for experiments conducted with the Phi-2-2.7B model. It is important to note that execution times may vary due to differences in computational resources and runtime conditions. To ensure the reliability of our measurements, repeated experiments are conducted in a stable environment, and the averaged statistics are reported below:

| Model | Total Execution Time | Normalized Execution Time |
|---|---|---|
| MagiCoder-S-DS-6.7B | 0.29 | 2.37 |
| After CodeDPO | 0.21 | 1.58 |

| Model | Total Max Memory Usage | Normalized Max Memory Usage |
|---|---|---|
| MagiCoder-S-DS-6.7B | 24.71 | 1 |
| After CodeDPO | 23.48 | 1 |

| Model | Total Memory Usage | Normalized Memory Usage |
|---|---|---|
| MagiCoder-S-DS-6.7B | 4.57 | 2.36 |
| After CodeDPO | 3.90 | 1.93 |

Table 9: Performance comparison on EffiBench for execution time and memory usage.

| Benchmark | Before CodeDPO (s) | After CodeDPO (s) | Average Speedup |
|---|---|---|---|
| HumanEval+ | 0.250 | 0.172 | 1.45x |
| MBPP+ | 0.189 | 0.137 | 1.38x |

Table 10: Average execution time and speedup with CodeDPO.

These results demonstrate the consistent improvements in execution efficiency achieved through CodeDPO, highlighting its practical benefits in reducing runtime.

## G   ABLATION ON SAMPLE NUMBER FOR CODE AND TEST GENERATION

The choice of the sample number and temperature, as described in Section 4.2, is guided by practical considerations to balance the diversity of sampled code solutions and test cases. These parameters are selected based on empirical observations and insights from prior work on data generation. To further investigate this, we conduct a series of ablation studies to evaluate the impact of varying sample numbers. Specifically, we tested sample numbers of 5, 15, and 50, with the experimental setup aligned with the design in Section 5.3.1. Table 11 presents the Spearman correlation between the self-validation score and the actual code accuracy on the HumanEval dataset, and then shows the performance of the Phi-2-2.7B model for varying sample numbers, evaluated on both the HumanEval and HumanEval+ benchmarks. Similar trends are observed for other models. The results suggest that using `sample_num=15` achieves a favorable trade-off between diversity and computational feasibility. While larger sample numbers provide marginal gains, they come with increased computational costs.

| Sample Number ($n$) | Spearman Correlation | | HumanEval (%) | HumanEval+ (%) |
|---|---|---|---|---|
| 5 | 0.7425 | | 54.88 | 49.39 |
| 15 | 0.8598 | | 57.32 | 51.83 |
| 50 | 0.8613 | | 57.90 | 51.83 |

Table 11: Spearman correlation and performance of Phi-2-2.7B for different sample numbers.

## H   DISCUSSION

### H.1   COMPARISON OF DATASET STATISTICS

Since some baselines have not released their datasets, we rely on statistics reported in their respective papers for comparison. Below is a summary of dataset sizes and the number of unique questions, as both metrics are important—greater diversity in unique questions generally leads to higher dataset quality.

| Method | Total Samples | Unique Questions |
|---|---|---|
| CodeDPO | 114k | 114k |
| PLUM | Up to 120k | Up to 1,500 |
| Code-Optimise | ∼100k (extended in our reproduction) | Up to 384 |

Table 12: Comparison of dataset sizes and unique questions across methods.

For SFT datasets, OSS-Instruct often combines multiple data sources. For example, models like MagiCoder-S-DS-6.7B and MagiCoder-S-CL-7B are trained using:

| SFT Dataset | Samples |
|---|---|
| Magicoder-OSS-Instruct | ∼75k |
| Magicoder-Evol-Instruct | ∼110k |
| Combined | Up to 185k |

Table 13: Supervised fine-tuning dataset statistics.

Based on comparisons with other related works, the dataset sizes of CodeDPO appear to be of the same order of magnitude. CodeDPO provides a significantly higher diversity in unique questions compared to baselines like PLUM and Code-Optimise, which heavily reuse prompts and have limited diversity despite similar sample sizes. This diversity ensures a more robust preference optimization process, which is a key advantage over existing approaches.

## H.2 OVERLAP AVOIDANCE WITH EXISTING BENCHMARKS

The seed dataset for CodeDPO was randomly selected from the open-source pretraining dataset *The Stack*, consisting of approximately 100k functions. This design explicitly considers data decontamination, since the seed dataset has already gone through rigorous data decontamination. It suggests that our dataset is unlikely to introduce additional data leakage beyond the seeds. To ensure quality, we applied a simple filtering process using tools like Tree-sitter and Pyright for static analysis and code formatting.

We intentionally avoided introducing any prior knowledge that might lead to significant overlap with evaluation benchmarks. We also implemented post-sampling data decontamination, similar to MagiCoder and StarCoder. However, given the already low overlap, this process only removed fewer than 30 samples. Thus, we can ensure that there is no risk of the dataset containing examples highly similar to the test sets.

To assess potential overlap for the final dataset with exisiting benchmarks, we followed the methodology used in MagiCoder. Specifically, we calculated the cosine similarity between HumanEval and the synthetic data generated by different methods. Below are the average similarity scores:

| Dataset | Avg Similarity Score |
|---|---|
| Self-Instruct | 0.169 |
| Evol-Instruct | 0.131 |
| OSS-Instruct | 0.105 |
| CodeDPO | 0.109 |

Table 14: Average similarity scores between datasets and HumanEval.

These results demonstrate that CodeDPO has a comparable or even lower overlap with HumanEval than most other widely used datasets, ensuring the validity and reliability of our evaluation.

### H.3 IMPLEMENTATION OF THE SELF-VALIDATION SCORES

#### H.3.1 PYTHON IMPLEMENTATION OF THE SELF-VALIDATION SCORES

To enhance the understanding of the proposed algorithm, we provide a Python implementation illustrating the calculation process for the case in Figure 2 (specifically, Step 2 in the figure). The code demonstrates the iterative calculation of self-validation scores using a simplified example.

---

**Python Implementation of the Self-Validation Scores in Figure 2**

```python
import numpy as np

# Example task-solution-test matrix
task_sol_test_matrix = [
    [[1, 1, 0],   # Code1: Test1, Test2
     [1, 0, 0],   # Code2: Test1
     [0, 0, 1]]   # Code3: Test3
]
task_sol_test_matrix = np.array(task_sol_test_matrix)

# Initialize solution and test scores (score=1)
sol_score = np.array([[1, 1, 1]])
test_score = np.array([[1, 1, 1]])

# Define iterative scoring function
def iter_step_page_rank(solution_scores_t_1, \
        test_scores_t_1, beta):
    test_scores_t = test_scores_t_1 * (1 - beta) + \
        np.einsum("PCT,PC->PT", task_sol_test_matrix, \
            solution_scores_t_1) * beta
    solution_scores_t = solution_scores_t_1 * (1 - beta) + \
        np.einsum("PCT,PT->PC", task_sol_test_matrix, \
            test_scores_t) * beta
    return solution_scores_t, test_scores_t

# Perform 2 iterations with beta = 0.5
for i in range(2):
    sol_score, test_score = \
        iter_step_page_rank(sol_score, test_score, 0.5)

# Output final scores
print(sol_score, test_score)
```

---

#### H.3.2 HANDLING WEAK TEST CASES

Our designed algorithm is robust. The self-validation scores can reflect the confidence of each code solutions and test cases through the iterative process. Notably, even in the presence of weak test cases (such as `assert True`), our method handles them robustly. We have carefully considered the impact of weak test cases in our design. We address this issue from two perspectives: ❶ **Natural Suppression of Weak Test Cases in Ranking:** Weak test cases are those that almost all code solutions pass. While they contribute to the overall scores of all code solutions, they do not affect the relative differences between code solutions in the ranking process. Since the ranking is based on score differences, weak test cases naturally have minimal impact on the ranking outcomes. ❷ **Filtering Identical or Close Scores:** Weak test cases can lead to highly similar scores for multiple code solutions after repeated score updates, diminishing the ability to differentiate between them. To address this, as described in Section 3.4, we implement a filtering mechanism that excludes samples with identical or near-identical ranking scores. This ensures that the influence of weak test cases is mitigated in the final dataset.

For example, assume we have 15 code solutions and 15 test cases generated by the model. ❶ If a weak test case, such as assert True, is passed by all 15 code samples, its score during each update step (as computed by Equation 1) will contribute equally to the scores of all code solutions. As a result, it does not alter the relative ranking of the code solutions. ❷ If all 15 test cases are similarly weak, the scores of the code solutions will converge to identical or near-identical values after several updates. To mitigate this, we apply a post-processing step (Section 3.4) to filter out such cases, ensuring the integrity of the final rankings. By addressing weak test cases through these mechanisms, our algorithm achieves robustness and maintains the reliability of its outputs, even in challenging scenarios.

## H.4 FUTURE WORK

**Limitations of Current Correctness Evaluation**    The test-case-driven functional correctness DPO is still not enough for code model. Current methods for evaluating correctness heavily rely on high-quality test cases or powerful models (e.g., GPT-4) to generate reliable outputs. To address these limitations, our paper introduces a self-validation data generation method that reduces dependency on such resources while maintaining robustness.

Because our method does not require high-quality test cases or strong external models, it is well-suited for scaling to larger datasets and can be applied to a wide range of code models. This scalability provides a foundation for improving correctness and efficiency across diverse code tasks.

**Incorporating Readability and Security**    Beyond correctness and efficiency, incorporating readability and security metrics into our extended CodeDPO framework is a natural extension: Metrics such as comment-to-code ratio, consistent variable naming, and adherence to coding style guides could be integrated into the preference learning process. For instance, LLMs could act as judges to evaluate readability alongside correctness. Techniques like static code analysis and detection of code smell and common vulnerabilities could help identify and penalize insecure patterns during data construction, contributing to safer code generation. We plan to explore these deeper alignment objectives in future work.

