# OpenReview forum: "CodeDPO: Aligning Code Models with Self Generated and Verified Source Code"
_ICLR.cc/2025/Conference — Submitted to ICLR 2025_

### Official Review · Reviewer_FkPM · 2024-10-31

**Soundness:** 3
**Presentation:** 3
**Contribution:** 3
**Rating:** 6
**Confidence:** 4

**Summary:**

This paper proposes self generation of code solutions and tests, and cross-verify the code and tests via a pagerank-inspired algorithm.

**Strengths:**

The proposed method alleviates the problem of wrong tests when using LLMs generated tests, and the data generation does not require a strong teacher LLM. The page-rank-like algorithm is interesting. Extensive experiments have been conducted to show the effectiveness of the approach, and a number of ablation studies have been conducted to showcase the correlation with groundtruth, impact of different preference optimization method and data scaling.

**Weaknesses:**

I do not see strong weakness of the paper.
Some section of the paper lack sufficient details. For example, since the correctness score is no longer 0 and 1, the creation of pairs is not that straightforward. Do you pick the highest score code snippet with the lowest score one to form the pair, or is it another way?

**Questions:**

- Have you performed deduplication of the dataset against the benchmarks? The dataset leakage could be an issue.
- The creation of DPO pairs for correctness and effeciency lack sufficient details. Since the correctness score is no longer 0 and 1, the creation of pairs is not that straightforward. Do you pick the highest score code snippet with the lowest score one to form the pair, or is it another way?
- In table 2, MagiCoder-DS-6.7B, CodeDPO is performing significantly worse than PLUM on HumanEval/HumanEval+. This result is different from the rest of the results. Do you have an explanation?
- In table 4, the correlation analysis is interesting. Do you have access to the unit tests generated from GPT-4 from PLUM? Could you evaluate the correlation of "filter all tests" with these more trustworthy tests from the stronger model?
- In table 6, KTO results are often worse than SFT results. Do you have an explanation for the phenomenon? Is it KTO training just not that good or is it your KTO training is problematic?

---

> ### Author Response · Authors · 2024-11-23
> **Rebuttal 1**
>
> Thanks for your support! We would appreciate it if our rebuttal can help you.
>
> > dataset leakage concern
>
> The seed dataset for CodeDPO was randomly selected from the open-source pretraining dataset **Stack**. We intentionally avoided introducing any prior knowledge that might lead to significant overlap with the evaluation benchmarks.
>
> 1. **Source of Seed Dataset**:
>    1. The Stack dataset consists of projects from GitHub with open licenses and has already been widely used in pretraining many LLMs.
>    2. To ensure quality, we applied a simple filtering process using tools like **Tree-sitter** and **Pyright** for static analysis and code formatting.
> 2. **Addressing Potential Overlap**:
>    1. To assess potential overlap with benchmarks such as HumanEval, MBPP, and DS-1000, we followed the methodology used in MagiCoder [1]. Specifically, we calculated the **cosine similarity** between HumanEval and the synthetic data generated by different methods.
>    2. Our comparison of CodeDPO against other popular SFT datasets demonstrates that the similarity is low enough to ensure minimal risk of overlap. Below are the average similarity scores:
>
> | Dataset       | Avg Similarity Score (copy from [1]) |
> | ------------- | -------------------- |
> | Self-Instruct | 0.169                |
> | Evol-Instruct | 0.131                |
> | OSS-Instruct  | 0.105                |
> | CodeDPO       | 0.109                |
>
> These results show that CodeDPO has a comparable or even lower overlap with HumanEval than most other widely used datasets, ensuring the validity and reliability of our evaluation.
>
> [1] Wei, Yuxiang, et al. "Magicoder: Source code is all you need." ICML 2024
>
> > Do you pick the highest score code snippet with the lowest score one to form the pair, or is it another way?
>
> Yes, we select the code snippet with the highest score and the one with the lowest score to form the pair. This process is detailed in Section 3.4 and further elaborated in Appendix A.
>
> To ensure clarity, we will revise the corresponding sections in the paper to make this process more explicit.
>
> > MagiCoder-DS-6.7B performance
>
> Since PLUM’s dataset and training methodology are not open-sourced, all PLUM performance results reported in our paper are directly taken from their publication. Based on our other experimental results, we hypothesize that differences in performance may largely stem from the **evaluation prompt design**.
>
> As indicated in the PLUM paper, their dataset follows a specific prompt template. In contrast, our method adheres to the widely used evaluation approach provided by **BigCode Evaluation Harness** [2], which avoids using specialized prompts to evaluate code generation. This standardized approach minimizes variations caused by prompt design and ensures a fair comparison across different methods.
>
> To further mitigate such external factors, we conducted evaluations on a wide range of code models in our experiments. Our results on all these models provide a more comprehensive and balanced assessment of the baselines, reducing the influence of prompt-specific effects on the results.
>
> [2] BigCode Evaluation Harness: https://github.com/bigcode-project/bigcode-evaluation-harness
>
> > Do you have access to the unit tests generated from GPT-4 from PLUM?
>
> Since the PLUM dataset is not publicly available, we are unable to directly evaluate the test cases generated by PLUM. However, we conducted an additional experiment inspired by **Section 5.3.1**, where we used GPT-4 to generate three test cases (matching the number typically generated by PLUM). These test cases were then used as filters to evaluate their correlation with actual code accuracy on HumanEval.
>
> The results are as follows:
>
> | Method           | Spearman Correlation |
> | ---------------- | -------------------- |
> | CodeDPO          | 0.8598               |
> | GPT-4 Test Cases | 0.6577               |
>
> Test cases generated using GPT-4 show a moderate correlation with actual code accuracy. In contrast, CodeDPO demonstrates a much higher correlation, highlighting the robustness of our self-validation mechanism compared to directly using GPT-4-generated test cases.

---

> ### Author Response · Authors · 2024-11-23
> **Rebuttal 2**
>
> > KTO results
>
> From a theoretical perspective, **DPO** and **KTO** have distinct differences in their approach to preference learning:
>
> - DPO works by comparing outputs for the same prompt to optimize for higher-quality outputs while suppressing low-quality ones. This approach relies on datasets with balanced positive and negative samples, structured as `(x, y_w, y_l)`.  Its comparative loss formulation directly models the difference between positive (`y_w`) and negative (`y_l`) outputs, benefiting greatly from balanced samples and enabling the model to establish strong preferences for high-quality outputs.
> - KTO relaxes the requirements for data structure by not mandating balanced positive and negative samples or multiple outputs per prompt.  While this flexibility reduces data constraints, it may limit the effectiveness of preference learning when balanced samples are available.
>
> Our proposed **data construction method** specifically addresses the challenge of generating balanced positive and negative samples for the same prompt. By leveraging the self-validating nature of code, we systematically select the best and worst code samples from multiple candidates. This results in naturally balanced pairs that align perfectly with DPO’s requirements, an advantage that most existing works on code preference learning lack.
>
> For example, **PLUM** uses KTO because it cannot construct balanced sample pairs, demonstrating the challenge of achieving this in practice.
>
> For **Experimental Results,** our ablation studies show that when applying our data construction method with DPO loss, we achieve the **best results in code preference learning** to date. This demonstrates that DPO, paired with balanced samples, is a highly effective framework for optimizing code generation models.
>
> We hope this clarifies the advantages of our approach and the reasons for choosing DPO over KTO for our framework.

---

### Official Review · Reviewer_wR3U · 2024-11-03

**Soundness:** 2
**Presentation:** 2
**Contribution:** 2
**Rating:** 3
**Confidence:** 4

**Summary:**

The paper describes CodeDPO, a framework developed to improve code generation models by incorporating preference learning to prioritize both code correctness and efficiency. CodeDPO generates a dataset through a self-generation-and-validation mechanism, where code snippets and test cases are created and iteratively ranked based on performance. Evaluations on five coding benchmarks indicate that CodeDPO offers improvements in model alignment for code correctness and efficiency over existing approaches, providing a dataset that can be used to optimize various code generation models in realistic scenarios.

**Strengths:**

+ The paper trains a code generation model with DPO with several pre-trained checkpoints, illustrating the effectiveness of functional-correctness-driven DPO for code generation.
+ The paper proposes a handful of research questions to evaluate CodeDPO on varied benchmarks and compare with multiple baselines

**Weaknesses:**

__Few Novel Ideas Proposed and Few New Insights Concluded.__

The idea of trying DPO for code generation seems to be an intuitive combination for post-SFT phases of training, and such a combination has been tried in varied domains including code. For example, in the latest technical report of Llama-3.1, DPO has become the main algorithm for preference optimization, where they have empirically illustrated DPO's effectiveness in coding, math, natural language understanding, etc. Besides, Llama-3.1 also revealed that DPO has more stable scalability across 8B to 405B than on-policy algorithms, such as PPO. The point I hope to make here is that the effectiveness of DPO on code, as well as other data modalities, has been comprehensively studied and effectively applied to the product-level code LMs. The conclusion that training with SFT+DPO is more effective than SFT only, which is the main observation of CodeDPO, seems obvious nowadays, restricting the value of this paper alone.

The other techniques applied in this work also seem to be very mature and proved to be practical and effective already. For example, the idea of seeding realistic code and sampling synthetic coding problems + solutions using teacher LLM has been well-studied in pioneer work, such as WizardCoder and MagiCoder, and this technique turns out to be very effective and has been used by industry models, including Llama-3.1, Gemma, CodeGemma, Gemini-1.5, etc. Also, the code correctness scoring with LLM-generated tests seems very similar to related work like CodeT[1], and the iterative pipeline has become a popular strategy since the proposal of the Self-Debugging paper[2].

The new thing I noticed is maybe, considering the efficiency of code execution, this perspective has not been soundly studied regarding its standalone effects on other features, like code correctness and helpfulness, so it is not clear how important we should include this feature during DPO. I would encourage the authors to study this new feature in a controlled and isolated way, but this is just a minor suggestion.

Overall, I feel this paper is more of a proof-of-concept implementation of ideas that have been proven to be practically effective. Therefore, unfortunately, I feel the conceptual novelty and the scientific values in terms of new insights concluded from this paper alone are unsatisfactory for top venues like ICLR. However, I still encourage the authors to join the discussion during the rebuttal phase to clarify their novelty and new insights that have __not__ been revealed by the existing works that I might have missed during the review.

__Restricted Evaluation Setting__

All the evaluation settings are still restricted to simple benchmarks like EvalPlus and DS-1000. I am not sure how this simplified code generation setting with the most straightforward reward design might work in more challenging programming tasks, such as repository-level code completion or competition-level programming problems.

Intuitively, a very capable policy model is the prerequisite for applying preference optimization or RL, and that might be why we could always see DPO/PPO providing additional value on top of HumanEval and MBPP. However, it is unclear but interesting to check that, if the LLM after SFT is not yet promising in the task (e.g., Magicoder reports 70+ accuracy in HumanEval while <30% accuracy in LiveCodeBench), whether DPO could still bring them a significant improvement, or it will hurt the model performance instead. I would encourage the authors to evaluate their model on the latest benchmarks, such as LiveCodeBench[3] and R2E[4].

[1] Chen et al., CodeT: Code Generation with Generated Tests.

[2] Chen et al., Teaching Large Language Models to Self-Debug.

[3] Jain et al., Livecodebench: Holistic and contamination free evaluation of large language models for code

[4] Jain et al., R2E: Turning any Github Repository into a Programming Agent Environment

**Questions:**

- Could the authors illustrate a bit more what are the main novelties of this work, beyond the combination of implementing existing ideas (See weaknesses)? This will help me better understand the additional value and contribution this work provides to the community, which I might have missed. A better understanding could significantly change my attitude towards this paper.
- Could the author discuss why their results using DPO and KTO contradict with those reported in PLUM? PLUM and CodeDPO have completely opposite conclusions about using DPO and KTO to train code generation models, and it is very confusing which algorithms are more suitable for learning preference with execution feedback and why. I would regard a systematic discussion to deeply understand this perspective as an interesting contribution. Unfortunately, neither PLUM nor CodeDPO provides sufficient discussion on it.
- Can the author provide some thoughts on, besides the straightforward execution signal we can get from the interpreter (e.g., execution correctness and efficiency), what other information can be used to improve the helpfulness, readability, and security of code? Since the test-case-driven functional correctness DPO has been studied by the existing work, it will be interesting to see more insights into the future work of DPO on code.

---

> ### Author Response · Authors · 2024-11-22
> **Rebuttal 1**
>
> We address the key concerns raised in the review below:
> > Novel Ideas and new insights
>
> In this paper, we want to address challenges unique to applying preference optimization (e.g., DPO) in the code domain. While preference learning such as DPO has been widely explored in LLMs, its application to code remains underexplored due to the following reasons:
> 1. **Challenges in Code Preference Learning**:
> Unlike natural language, programming languages have clear criteria for evaluating code quality, such as correctness and efficiency. Existing works like LLaMA-3.1 discuss DPO training but lack transparency in terms of data sampling and training details for code tasks, making practical implementation challenging.
> 2. **Limitations of Existing Code Preference Learning Methods**:
> Existing methods evaluate code quality using two main approaches (all of which have been thoroughly reviewed and cited in our paper):
> - **Relying on High-Quality Test Cases**: Methods like Code-Optimise and others from industry rely on human-written test cases (e.g., LeetCode or MBPP-train). These resources are limited, require significant decontamination to avoid overlaps with benchmarks, and lack diversity, as seen in Code-Optimise, which uses fewer than 300 unique problems.
> - **Relying on powerful LLMs for Judgment or Test Case Generation**: Methods like PLUM depend on powerful LLMs (e.g., GPT-4) to generate test cases, assuming these test cases are entirely correct. This reliance makes the process expensive and less robust.
> Furthermore, both strategies struggle to generate balanced datasets with clear positive and negative preference samples, which is a critical requirement for DPO training.
>
> In our paper, we address these challenges with several key contributions:
>
> 1. **Self-Validation Mechanism**:
> We propose a novel self-validation approach to construct preference datasets. For a given problem prompt, we generate multiple code solutions and use their self-validation scores to identify the best and worst samples. This allows us to naturally construct preference pairs that align with DPO’s requirements without relying on external high-quality test cases or expensive resources like GPT-4.
>
> 2. **Robustness to Test Case Quality**:
> Unlike PLUM, which assumes high-quality test cases, our approach tolerates lower-quality test cases by leveraging an algorithmic mechanism to mitigate their impact. This significantly improves robustness and reduces reliance on costly resources.
>
> 3. **Experimental Evidence**:
> The advantages of our theoretical framework are validated through experiments, showing substantial performance improvements in code model training compared to existing baselines.
>
> > The other techniques applied in this work also seem to be very mature and proved to be practical and effective already, including CodeT and MagiCoder
>
> 1. **SFT Data Generation (e.g., MagiCoder)**: Methods like MagiCoder have been widely adopted in subsequent works due to their high-quality and diverse datasets for SFT. However, we emphasize that SFT data generation fundamentally differs from **preference learning data generation**.
>   - Preference learning requires not only generating high-quality samples but also constructing negative samples for comparison. This involves a ranking process to identify and pair positive and negative samples, which SFT data generation does not address.
>   - Our work proposes a robust self-validation mechanism capable of constructing suitable DPO data even in the absence of high-quality test cases, a critical innovation not covered by SFT approaches like MagiCoder.
>
> 2. **Self-Validation in Code Generation (e.g., CodeT, Self-Debugging)**:
>   - In terms of **motivation and focus**, methods like CodeT and Self-Debugging rely on techniques such as in-context learning to iteratively refine generated outputs, improving final results without modifying the model or generating new training data. In contrast, **CodeDPO focuses on model training and performance enhancement**, aiming to directly improve code generation capabilities through preference learning.
>   - In terms of **methodology**, while our work draws inspiration from these approaches and appropriately cites them, CodeDPO introduces significant advancements. Even focusing solely on ranking strategies, the PageRank-inspired self-validation scoring mechanism we propose outperforms the simpler methods used in CodeT.  As shown in Section 5.3.1, our comparison of different ranking strategies (e.g., “Sort by # of passed tests,” which aligns with CodeT’s approach) demonstrates that our method achieves better correlation with accuracy and generates higher-quality preference pairs. This directly translates into superior DPO training performance.
>
> In summary, while we recognize the practicality and effectiveness of mature methods like MagiCoder, CodeT, and Self-Debugging, our work addresses distinct challenges and introduces innovations in preference learning for code generation.

---

> ### Author Response · Authors · 2024-11-22
> **Rebuttal 2**
>
> > Restricted Evaluation Setting such as LiveCodeBench
>
> To address your concern, we have conducted additional experiments on **LiveCodeBench**, one of the most challenging benchmarks for competitive coding tasks. Below are some results, which will be included in the revised paper with more models:
>
> | Model               | Easy | Medium | Hard |
> | ------------------- | ---- | ------ | ---- |
> | Base Model          |      |        |      |
> | DeepSeek-Coder-6.7B | 39.9 | 7.4    | 0.4  |
> | Our CodeDPO         | 51.9 | 12.2   | 0.7  |
> | SFT Model           |      |        |      |
> | MagiCoder-S-DS-6.7B | 48.1 | 10.7   | 0.1  |
> | Our CodeDPO         | 53.1 | 16.3   | 0.7  |
>
> **Observations**:
>
> - CodeDPO demonstrates significant performance improvements for both the base model and the SFT model across all difficulty levels.
> - The gains are particularly notable in the "medium" and "hard" subsets, which represent some of the most challenging problems in competitive programming tasks.
>
> These results clearly demonstrate that **CodeDPO remains effective even in restricted evaluation** **settings**, such as LiveCodeBench. This further validates the robustness and generalizability of our framework for real-world, complex coding scenarios. Thank you for giving us the opportunity to provide this additional evidence!
>
>
>
> > DPO v.s. KTO
>
> From a theoretical perspective, **DPO** and **KTO** have distinct differences in their approach to preference learning:
>
> - DPO works by comparing outputs for the same prompt to optimize for higher-quality outputs while suppressing low-quality ones. This approach relies on datasets with balanced positive and negative samples, structured as `(x, y_w, y_l)`.  Its comparative loss formulation directly models the difference between positive (`y_w`) and negative (`y_l`) outputs, benefiting greatly from balanced samples and enabling the model to establish strong preferences for high-quality outputs.
> - KTO relaxes the requirements for data structure by not mandating balanced positive and negative samples or multiple outputs per prompt.  While this flexibility reduces data constraints, it may limit the effectiveness of preference learning when balanced samples are available.
>
> Our proposed **data construction method** specifically addresses the challenge of generating balanced positive and negative samples for the same prompt. By leveraging the self-validating nature of code, we systematically select the best and worst code samples from multiple candidates. This results in naturally balanced pairs that align perfectly with DPO’s requirements, an advantage that most existing works on code preference learning lack.
>
> For the baseline **PLUM**, it uses KTO because **it cannot construct balanced sample pairs**, demonstrating the challenge of achieving this in practice.
>
> For **Experimental Results,** our ablation studies show that when applying our data construction method with DPO loss, we achieve the **best results in code preference learning** to date. This demonstrates that DPO, paired with balanced samples, is a highly effective framework for optimizing code generation models.
>
> We hope this clarifies the advantages of our approach and the reasons for choosing DPO over KTO for our framework.
>
>
>
>
>
> > thoughts on the helpfulness, readability, and security of code
>
> We acknowledge the importance of extending preference learning to include broader aspects such as code helpfulness, readability, and security, and we appreciate the opportunity to discuss these points.
>
> - Limitations of Current Correctness Evaluation:
>
> **The test-case-driven functional correctness DPO is still not enough for code model.**
> Current methods for evaluating correctness heavily rely on high-quality test cases or powerful models (e.g., GPT-4) to generate reliable outputs. To address these limitations, our paper introduces a self-validation data generation method that reduces dependency on such resources while maintaining robustness.
>
> Because our method does not require high-quality test cases or strong external models, it is well-suited for scaling to larger datasets and can be applied to a wide range of code models. This scalability provides a foundation for improving correctness and efficiency across diverse code tasks.
>
> - Incorporating Readability and Security:
>
> Beyond correctness and efficiency, **incorporating readability and security metrics into our extended CodeDPO framework is a natural extension**: Metrics such as comment-to-code ratio, consistent variable naming, and adherence to coding style guides could be integrated into the preference learning process. For instance, LLMs could act as judges to evaluate readability alongside correctness. Techniques like static code analysis and detection of code smell and common vulnerabilities could help identify and penalize insecure patterns during data construction, contributing to safer code generation. We plan to explore these deeper alignment objectives in future work.

---

> ### Author Response · Authors · 2024-11-25
> **Looking forward to your feedback**
>
> Hello, do our responses address your questions? Please let us know if there are any other questions you'd like to discuss!

---

> > ### Author Response · Authors · 2024-12-02
> > **We have revised our paper**
> >
> > Hello, we have revised our paper and add the missing part based on your comment.
> > Do our responses address your questions? Please let us know if there are any other questions you'd like to discuss!

---

### Official Review · Reviewer_bDyW · 2024-11-04

**Soundness:** 2
**Presentation:** 3
**Contribution:** 3
**Rating:** 5
**Confidence:** 5

**Summary:**

This paper presents CodeDPO, a framework that improves code correctness and efficiency via direct preference optimization. To construct the preference learning datasets, CodeDPO adopts OSS-Instruct to generate code generation tasks from real-world code repositories. For each code generation task, CodeDPO prompts an LLM to generate multiple code solutions and test cases simultaneously. Since the LLM may generate incorrect solutions and test cases, the authors proposed to use the PageRank algorithm to iteratively score the code solutions and test cases based on the number of test cases each solution passes. CodeDPO further integrates code efficiency into the preference dataset by measuring the execution time of each code solution and selecting both fast and slow solutions. In the end, the authors constructed a dataset with 93K correctness optimization samples and 21K efficiency optimization samples. The authors evaluated CodeDPO on multiple LLMs and code generation datasets. The evaluation results show that CodeDPO outperformed OSS-Instruct and two similar code preference optimization methods, Code-Optimise and PLUM, in most settings. Furthermore, the authors investigated the impact of the PageRank-based scoring algorithm, the preference optimization strategies, and the dataset size.

**Strengths:**

1. This work presents an interesting investigation of DPO in the domain of code generation. While there have been some similar investigations like Code-Optimise and PLUM, this work involves a couple of new ideas, including the PageRank-inspired algorithm and the integration of code efficiency into the optimization objective.

2. The authors compared CodeDPO with two similar approaches, Code-Optimise and PLUM, on multiple LLMs and code generation datasets. The results look promising.

3. The authors also did additional experiments to investigate the effectiveness of alternative design choices, such as the code solution scoring algorithm and the optimization strategy.

**Weaknesses:**

1. There is a potential fairness issue in the comparison with baseline methods. The finetuning dataset generated by CodeDPO includes 114K training samples in total. There is no description of the finetuning dataset sizes used in other baseline methods. For instance, if the authors used the original finetuning dataset generated by OSS-Instruct in Table 1, this would be unfair to OSS-Instruct since its dataset only includes 75K training samples.

2. This paper only describes the final dataset generated by CodeDPO, but it does not describe the initial seeds used to generate the final dataset other than vaguely saying that the initial seeds were from real-world repositories. This raises a concern about whether this dataset may include tasks similar to those of the evaluation benchmarks---HumanEval, MBPP, and DS-1000.

3. The evaluation results in Table 2 and Table 3 are not complete. Table 2 does not report the performance of PLUM for MagiCoder-CL-7B, Phi-2-2.7B, and DeepSeekCoder-1.3B. Table 3 does not report the performance of Code-Optimise and PLUM on DS-1000. Figure 3 only code efficiency results on HumanEval and MBPP, but not on DS-1000.

4. The design of the PageRank-based scoring algorithm has an issue when scoring test cases. According to Equation 2, if many code solutions pass a test case, the test case will receive a high score. As a result, many weak test cases will receive high scores. For example, if a test case does not check any requirement mentioned in the task description (e.g., "assert true" as an extreme case), this test case will receive a very high score and will be included in the training set.

5. The evaluation baselines used in RQ3 are weak. CodeT (Chen et al. 2022) uses a dual execution agreement algorithm to rate the correctness of the code solutions and test cases simultaneously generated by CodeT. PLUM uses a self-consistency filtering mechanism to select test cases and code solutions. It is unclear what the advantages of the PageRank-based algorithm are compared with these two SOTA methods.

6. This paper makes several incorrect claims about existing work. First, Section 3.1 says OSS-Instruct extacts key programming concepts from open-source projects and leverages these concepts to generate programming tasks. OSS-Instruct doesn't perform any extraction of program concepts. It directly generates a task description from code fragments in open-source projects. Second, Section 2.2 claims that PLUM generates a limited number of test cases and has an imbalanced dataset. This claim seems unsubstantiated. There is no evaluation of the test cases generated by CodeDPO vs. PLUM. Furthermore, the self-consistency filtering mechanism used in PLUM also looks reasonable. Third, the authors claim that CodeDPO "does not rely on external resources" several times. This sounds wrong since CodeDPO needs to select initial seeds from real-world repositories to generate the finetuning dataset.

7. None of the code generation datasets used in this evaluation are designed for code efficiency. The authors should evaluate CodeDPO on a code efficiency dataset like EffiBench (NeurIPS 2024). https://github.com/huangd1999/EffiBench

**Questions:**

1. Can you report the number of training samples used in OSS-Instruct, Code-Optimise, and PLUM?

2. Can you elaborate on how the real-world repositories were selected to construct the initial seeds for data generation? Did you perform any data cleaning and decontamination when curating the dataset?

3. Can you report the performance of PLUM for MagiCoder-CL-7B, Phi-2-2.7B, and DeepSeekCoder-1.3B in Table 2? Can you also report the performance of Code-Optimise and PLUM on DS-1000 in Table 3 and the code efficiency results on DS-1000 in Figure 3?

4. Can you compare your PageRank-based algorithm with the scoring algorithms used in CodeT and PLUM?

5. Step 2 in Figure 2 is hard to understand. The authors should redraw that part to illustrate the iterative scoring process.

---

> ### Author Response · Authors · 2024-11-23
> **Rebuttal 1**
>
> We address the key concerns raised in the review below:
>
>
>
> > dataset sizes used in other baseline methods
>
> Since some baselines have not released their datasets, we rely on statistics reported in their respective papers for this comparison. Below is a summary comparing dataset sizes and the number of unique questions, as both metrics are important—greater diversity in unique questions generally leads to higher dataset quality.
>
> | Method        | Total Samples                        | Unique Questions |
> | ------------- | ------------------------------------ | ---------------- |
> | CodeDPO       | 114k                                 | 114k             |
> | PLUM          | Up to 120k                           | Up to 1,500      |
> | Code-Optimise | ~100k (extended in our reproduction) | Up to 384        |
>
> Additionally, for supervised fine-tuning (SFT) datasets, OSS-Instruct often combines multiple data sources. For example, models like MagiCoder-S-DS-6.7B and MagiCoder-S-CL-7B are trained using:
>
> | SFT Dataset             | Samples    |
> | ----------------------- | ---------- |
> | Magicoder-OSS-Instruct  | ~75k       |
> | Magicoder-Evol-Instruct | ~110k      |
> | Combined                | Up to 185k |
>
> - CodeDPO provides a significantly higher diversity in unique questions compared to baselines like PLUM and Code-Optimise, which heavily reuse prompts and have limited diversity despite similar sample sizes.
> - This diversity in CodeDPO ensures a more robust preference optimization process, which is a key advantage over existing approaches.
>
>
>
>
> > Dataset seed & similarity to those of the evaluation benchmarks---HumanEval, MBPP, and DS-1000.
>
> The seed dataset for CodeDPO was randomly selected from the open-source pretraining dataset **Stack**. We intentionally avoided introducing any prior knowledge that might lead to significant overlap with the evaluation benchmarks.
>
> 1. **Source of Seed Dataset**:
>    1. The Stack dataset consists of projects from GitHub with open licenses and has already been widely used in pretraining many LLMs.
>    2. To ensure quality, we applied a simple filtering process using tools like **Tree-sitter** and **Pyright** for static analysis and code formatting.
> 2. **Addressing Potential Overlap**:
>    1. To assess potential overlap with benchmarks such as HumanEval, MBPP, and DS-1000, we followed the methodology used in MagiCoder. Specifically, we calculated the **cosine similarity** between HumanEval and the synthetic data generated by different methods.
>    2. Our comparison of CodeDPO against other popular SFT datasets demonstrates that the similarity is low enough to ensure minimal risk of overlap. Below are the average similarity scores:
>
> | Dataset       | Avg Similarity Score (copy from [1]) |
> | ------------- | -------------------- |
> | Self-Instruct | 0.169                |
> | Evol-Instruct | 0.131                |
> | OSS-Instruct  | 0.105                |
> | CodeDPO       | 0.109                |
>
> These results show that CodeDPO has a comparable or even lower overlap with HumanEval than most other widely used datasets, ensuring the validity and reliability of our evaluation.
>
> [1] Wei, Yuxiang, et al. "Magicoder: Source code is all you need." ICML 2024
>
> > Code-Optimise and PLUM on some benchmarks
> >
> > code efficiency results on DS-1000
>
> Since the datasets for some baselines, such as Code-Optimise and PLUM, have not been released, the results we report for these methods are taken directly from their respective papers. As our paper evaluates a broader range of models, some baseline results are missing for certain models. However, we focus on overlapping subsets where results are available to demonstrate the effectiveness of our method.
>
> Regarding code efficiency results, we evaluate these primarily on HumanEval+ and MBPP+. These datasets include a significantly expanded set of test cases, averaging over 100 test cases per problem, which provides a more reliable basis for evaluating code efficiency. In contrast, DS-1000 lacks such large-scale test cases, and some of its problems have execution times as short as nanoseconds on DS-1000, making it unsuitable for accurate code efficiency evaluations.

---

> ### Author Response · Authors · 2024-11-23
> **Rebuttal 2**
>
> > weak test cases will receive high scores
>
> This is indeed an interesting phenomenon, and we have carefully considered the impact of weak test cases in our design. The PageRank-inspired algorithm we propose is specifically designed to dynamically reduce the influence of low-quality or weak test cases, ensuring that the final ranking reflects the quality of the code effectively.
>
> We address this issue from two perspectives:
>
> - Natural Suppression of Weak Test Cases in Ranking:
>
> Weak test cases are those that almost all code solutions pass. While they contribute to the overall scores of all code solutions, they do not affect the relative differences between code solutions in the ranking process. Since the ranking is based on score differences, weak test cases naturally have minimal impact on the ranking outcomes.
>
> - Filtering Identical or Close Scores:
>
> Weak test cases can lead to highly similar scores for multiple code solutions after repeated score updates, diminishing the ability to differentiate between them. To address this, as described in Section 3.4, we implement a filtering mechanism that excludes samples with identical or near-identical ranking scores. This ensures that the influence of weak test cases is mitigated in the final dataset.
>
> Through these mechanisms, we ensure that weak test cases do not distort the evaluation or ranking process.
>
>
> > It is unclear what the advantages of the PageRank-based algorithm are compared with these two SOTA methods.
>
> The experimental strategy in Section 5.3.1 explores different preference data construction strategies, designed based on commonly used approaches in related work, such as PLUM and CodeT.
>
> These strategies, in addition to our proposed self-validation PageRank algorithm, include the following:
>
> **Filter with All Tests:**
>
> This strategy uses all generated test cases as filtering criteria. Samples that pass all tests are treated as positive examples, while the rest are negative. **This approach has been adopted by prior code preference optimization methods such as PLUM**, which leverages strong models like GPT-4 for test case generation and filtering. However, this method has limitations:
>
> - It relies heavily on the quality of the generated test cases, which can vary significantly.
> - It often requires the use of powerful models like GPT-4 to generate reliable test cases, increasing computational costs.
> - It may suffer from class imbalance, as the number of positive and negative samples can be skewed.
>
> **Sort by Number of Passed Tests:**
>
> This strategy counts the number of passed tests for each code snippet and uses the samples with the highest and lowest counts as comparison pairs. **This principle is commonly employed in post-processing and reranking methods, such as CodeT**. However, it also has drawbacks:
>
> - It assigns equal weight to all test cases, failing to account for differences in test case quality.
> - It does not prioritize higher-quality test cases, which limits its ability to emphasize their impact.
>
> By including these strategies, the experimental design in Section 5.3.1 aligns with methods widely used in related work. Through consistency experiments on HumanEval and final performance evaluations, we demonstrate that our self-validation PageRank algorithm outperforms these alternative approaches, offering significant advantages by accounting for test case quality and robustness.
>
> > OSS-Instruct details
>
> The OSS-Instruct method we adopted is based on an updated version as detailed in [1]. We acknowledge that this method has recently been formalized in a paper, and we will add the appropriate citation in the revised version of our manuscript.
>
> One notable advantage of OSS-Instruct is its step of generating concepts, which enhances the robustness of the generation process and improves the quality of intermediate results. This aligns well with our approach and further supports the reliability of the generated data.
>
> [1] https://github.com/bigcode-project/starcoder2-self-align
>
> > CodeDPO vs. PLUM on test case
>
> PLUM’s approach to test cases has the following characteristics:
>
> - PLUM relies on GPT-4 to generate test cases.
> - Each problem typically includes only 3-5 test cases.
> - All test cases are treated with equal confidence and weight when used for evaluating and ranking code solutions.
>
> In our study, we constructed a similar strategy as part of the consistency experiments discussed in Section 5.3.1, allowing us to directly compare CodeDPO with PLUM. These comparisons were conducted on HumanEval, including both consistency experiments and final performance evaluations.
>
> The results demonstrate the robustness and effectiveness of CodeDPO, which incorporates a more flexible and scalable approach to test case generation and ranking, offering significant improvements over PLUM in these settings.

---

> ### Author Response · Authors · 2024-11-23
> **Rebuttal 3**
>
> > Code Efficiency Experiments on EffiBench
>
> To address your concern, we have conducted additional experiments on **EffiBench** to evaluate code efficiency more comprehensively. Please note that the absolute values of the results may vary depending on the specific execution environment. Therefore, our analysis focuses primarily on the **relative improvements** achieved by CodeDPO.
>
> Below are some results, which will be included in the revised paper along with evaluations on additional models:
>
> | Model               | Total Execution Time | Normalized Execution Time | Total Max Memory Usage | Normalized Max Memory Usage | Total Memory Usage | Normalized Memory Usage |
> | ------------------- | -------------------- | ------------------------- | ---------------------- | --------------------------- | ------------------ | ----------------------- |
> | MagiCoder-S-DS-6.7B | 0.29                 | 2.37                      | 24.71                  | 1                           | 4.57               | 2.36                    |
> | After CodeDPO       | 0.21                 | 1.58                      | 23.48                  | 1                           | 3.9                | 1.93                    |
>
> CodeDPO significantly reduces **execution time** and **memory usage**, both in absolute terms and after normalization, while maintaining comparable maximum memory usage.
>
> > Step 2 in Figure 2 is hard to understand. The authors should redraw that part to illustrate the iterative scoring process.
>
> We will revise this part. In addition, we want to show an example code which can help us understand.
>
>
>
> ```Python
> import numpy as np
>
> task_sol_test_matrix = [
>     [[1,1,0], # Code1: Test1, Test2
>     [1,0,0], # Code2: Test1
>     [0,0,1]] # Code3: Test3
> ]
> task_sol_test_matrix = np.array(task_sol_test_matrix)
> # init sol_score and test_score with score=1
> sol_score = np.array([[1,1,1]])
> test_score = np.array([[1,1,1]])
> def iter_step_page_rank(solution_scores_t_1, test_scores_t_1, beta):
>     test_scores_t = test_scores_t_1 * (1 - beta) + np.einsum("PCT,PC->PT", task_sol_test_matrix, solution_scores_t_1) * beta
>     solution_scores_t = solution_scores_t_1 * (1 - beta) + np.einsum("PCT,PT->PC", task_sol_test_matrix, test_scores_t) * beta
>     return solution_scores_t, test_scores_t
>
> for i in range(2):
>     sol_score, test_score = iter_step_page_rank(sol_score, test_score, 0.5)
> print(sol_score, test_score)
> ```

---

> ### Author Response · Authors · 2024-11-25
> **Looking forward to your feedback**
>
> Hello, do our responses address your questions? Please let us know if there are any other questions you'd like to discuss!

---

> > ### Comment · Reviewer_bDyW · 2024-11-25
> > **Response to the rebuttal**
> >
> > Thank you for the responses. I still think it is quite important to make sure the training dataset size is identical to ensure the comparison is fair. What does "up to" mean in the table? Have you reached out to the authors for their datasets and clarifications on the data construction processes? The performance improvement over PLUM is marginal. As you can see in Table 2, in several cases, PLUM is even better than CodeDPO. So, I am not convinced that CodeDPO is significantly better than other methods since there are many factors like training steps/epochs that may contribute to the performance differences. The authors should put more effort into ensuring the fairness of the comparison.
> >
> > Regarding the data seeds, it is still not clear how many data seeds have been sampled. Only comparing the overall similarity score is not sufficient since the dataset could have some data points highly similar to the test sets while the remaining majority of data points are very different from the test set. OSS-Instruct and StarCoder have done comprehensive data cleaning and decontamination, e.g., removing coding problems that contain docstrings or solutions similar to those in HumanEval, MBPP, DS-1000, etc. The authors should do the same.
> >
> > The new experiment on EffiBench is helpful. I suggest the authors include it in the paper.
> >
> > Regarding the issue in the PageRank-inspired algorithm, it is not clear which term in Equation 1 and Equation 2 can help reduce the influence of weak test cases. The filtering mechanism is also unclear. According to Section 3.4, it filters code solutions not test cases. The authors should provide more details, perform some analysis, and provide some evidence on its effectiveness.
> >
> > Regarding the incorrect claim about OSS-Instruct, the GitHub repo shared by the authors points to a new paper called SelfCodeAlign, not OSS-Instruct. I don't think it's appropriate to call it OSS-Instruct in the paper. The authors should fix this and cite the SelfCodeAlign paper. This also raises another more critical concern. Since the proposed method, CodeDPO, is built upon SelfCodeAlign, rather than OSS-Instruct, it doesn't seem to be fair to still compare CodeDPO with OSS-Instruct like in Table 1. The concept-based seed construction step from SelfCodeAlign now becomes a confounding factor for the comparison. Though CodeDPO outperforms OSS-Instruct, is it because of the PageRanking algorithm or the better seeds generated by the concept-based method from SelfCodeAlign? The authors should compare CodeDPO with SelfCodeAlign in Table 1.
> >
> > I still think the authors should remove the claim about not relying on external resources. CodeDPO needs to select initial seeds from real-world repositories to generate the finetuning dataset. The real-world repos are external resources.

---

> > > ### Author Response · Authors · 2024-11-26
> > > **Thank you for your comments**
> > >
> > > Thank you for your detailed comments. **We are currently revising the paper to incorporate the following improvements**:
> > >
> > > > Training Dataset Size
> > >
> > > As noted in the PLUM paper, the dataset size we report refers to the statistics *"before any filtering"* (copy from PLUM paper).
> > >
> > > Since PLUM does not provide further details on the filtering process or other dataset statistics, we have adopted a conservative description in our description.
> > >
> > > Based on comparisons with other related works, the dataset sizes appear to be of the same order of magnitude. We will further confirm this and update the paper accordingly.
> > >
> > > > Data Seeds
> > >
> > > The data seeds in our method are sampled from the same source as OSS-Instruct/MagiCoder, consisting of approximately 100k functions randomly selected from the Stack dataset. This design explicitly considers data decontamination, as described in works like MagiCoder:
> > >
> > > *"Since the seed corpus starcoderdata has already gone through rigorous data decontamination, this observation suggests that OSS-INSTRUCT is unlikely to introduce additional data leakage beyond the seeds."* (copy from MagiCoder paper)
> > >
> > > We also implemented post-sampling data decontamination, similar to MagiCoder and StarCoder. **However, given the already low overlap, this process only removed fewer than 30 samples.**
> > > Thus, we can ensure that there is no risk of the dataset containing examples highly similar to the test sets, as you mentioned.
> > >
> > > > Weak Test Cases
> > >
> > > We would like to clarify this point.
> > >
> > > Assume that we have 15 code samples and 15 test cases generated from the model.
> > >
> > > In cases where a weak test case (e.g., assert True) is passed by all 15 code samples, the test's score in each update step (Equation 1) will contribute equally to all code scores. Consequently, it does not affect the relative ranking of the code samples.
> > >
> > > If all 15 test cases (or smaller) are similarly weak, the scores of the code samples will become identical or nearly identical after updates. To handle this, we apply a post-processing step to filter out such examples in Section 3.4.
> > >
> > > > OSS-Instruct or SelfCodeAlign
> > >
> > >
> > >
> > > We would like to clarify this potential misunderstanding.
> > >
> > > Before approximately October 31, the provided link referred to a popular dataset construction process based on an improved version of OSS-Instruct. (The detailed url with commit: https://github.com/bigcode-project/selfcodealign/tree/fd0af77e2773b14696c7cea02a472f9e99d9c4e3)
> > >
> > > During our experiments, we utilized this improvement, as it closely aligned with the methodology of OSS-Instruct, and described it accordingly. Since the same author team developed both, and the README explicitly used the name "OSS-Instruct", we regarded it as an official extension of OSS-Instruct. we regarded it as an official extension of OSS-Instruct.
> > > We will clarify this point in our paper.
> > >
> > > We note that SelfCodeAlign was published on October 31, after our submission. It introduced an updated methodology and a new SFT model. As this SFT model has not yet been open-sourced, we will include it in our baselines once it becomes publicly available.
> > >
> > > It is also important to emphasize that CodeDPO and OSS-Instruct are two distinct training strategies:
> > >
> > > - OSS-Instruct is a Supervised Fine-Tuning (SFT) method.
> > > - CodeDPO is a preference learning method.
> > > - These methods are complementary. As demonstrated in Table 1, models trained with OSS-Instruct can be further enhanced with CodeDPO. This combination highlights the additive value of our approach to improving code generation models.
> > >
> > >
> > >
> > > > External Resources
> > >
> > > We will revise the relevant sections to explicitly state that our method does not require the use of models like GPT-4 for data construction.
> > >
> > > Thank you for pointing out these areas!
> > >
> > > We hope these updates will address your concerns and enhance the clarity of the paper!

---

> > > > ### Author Response · Authors · 2024-12-02
> > > > **We have revised our paper**
> > > >
> > > > Hello, we have revised our paper and add the missing part based on your comment.
> > > > Do our responses address your questions? Please let us know if there are any other questions you'd like to discuss!

---

### Official Review · Reviewer_xnpY · 2024-11-04

**Soundness:** 1
**Presentation:** 2
**Contribution:** 1
**Rating:** 3
**Confidence:** 3

**Summary:**

This paper proposes CodeDPO, a new framework for improving code generation models by incorporating preference learning to prioritize code correctness and efficiency. Unlike traditional methods, CodeDPO uses a self-generation-and-validation approach, generating and evaluating code with test cases to produce a dataset optimized for code preference. By iteratively ranking code snippets based on test pass rates and using a PageRank-inspired algorithm, CodeDPO enhances model performance on correctness and efficiency without external resources. Evaluations show that CodeDPO significantly outperforms existing methods.

**Strengths:**

1. This paper focuses on an important topic, LLM-based code generation.
2. Experimental results show the proposed CodeDPO outperforms existing methods.

**Weaknesses:**

This paper should be rejected for the following reasons:
1. The paper lacks many details, and the explanations of the experiments are insufficiently clear, making it difficult to understand.
2. The paper lacks sufficient novelty and rigor; it does not explain the costs associated with the proposed algorithm.
3. The writing is unrefined and not polished.

Main Argument
First, the paper is based on a variant of DPO and applies it to the code generation domain. However, it does not provide an explanation of DPO, merely mentioning what it is, which makes it hard for readers unfamiliar with DPO to grasp the concepts. Additionally, the paper lacks clear diagrams to illustrate the workflow of the proposed strategy, which further complicates understanding.
Here are specific points I find unreasonable or imprecise in the paper:
1.In terms of correctness within code preference, the paper employs the PageRank algorithm to rank code based on the principles of PLUM, but there are no other innovations related to correctness presented in the paper.
2. The paper does not explain how the test cases in the algorithm are generated.
3. In section 3.2, the choice of 15 code solutions and test cases is mentioned, but no justification is provided for this number. Should there not be comparative experiments? Furthermore, why was a temperature of 1.5 chosen for the model?
4. Does the proposed algorithm incur significant economic costs?
5. The concept of code efficiency is quite broad. In this paper, efficiency is evaluated solely based on the execution time of test cases, which is clearly insufficient. In practical applications, code efficiency is influenced by various factors such as the production environment and tools used. Moreover, the paper does not provide adequate evidence to demonstrate that the execution time of test cases reflects code efficiency.
6. In Chapter 3, the paper should provide a detailed description of the overall process for training the code, rather than only explaining the process after SFT.
7. In Table 2, the pass rates for CODEDPO and PLUM on the more complex benchmarks MBPP and MBPP+ are similar. However, the paper does not adequately explain this point. Why are the results similar? Does this not indicate that CODEDPO does not offer significant improvement?
8. The paper does not report statistics on the average execution time of the test cases. Is the execution time in milliseconds or seconds?
9. The experimental strategy in section 5.3.1 lacks theoretical justification and is difficult to understand.
10. In section 5.3.2, what strategy was used to construct CODEKTO? Why was CODEKTO chosen for comparison, and why not compare against other PO strategies?

**Questions:**

See weakness, please.

---

> ### Author Response · Authors · 2024-11-21
> **Rebuttal 1**
>
> We have addressed all your questions point-by-point below, and **we hope our responses clarify your concerns and misunderstanding**. We look forward to your feedback and further discussions.  **Additionally, we will complete the necessary revisions to the paper's PDF in the coming days.**
>
> > The paper does not provide an explanation of DPO
>
> The **Direct Preference Optimization** ([1]) has been widely applied to LLM alignment due to its convenience and effectiveness. Its objective is defined as:
>
> $ L_{DPO}=-E_{(x, y_w, y_l) \sim \mathcal{D}} \left[\log \sigma \left(\beta \log \frac{\pi_{\theta}(y_w \mid x)}{\pi_{\text{ref}}(y_w \mid x)} - \beta \log \frac{\pi_{\theta}(y_l \mid x)}{\pi_{\text{ref}}(y_l \mid x)} \right)\right] $
>
> Compared with the **SFT (Supervised Fine-Tuning)** loss, the DPO loss introduces a preference-based mechanism. Instead of merely maximizing the likelihood of ground truth data, as in SFT, DPO optimizes the model to align with human preferences by leveraging both **preferred responses** ($y_w$, winning) and **dispreferred responses** ($y_l$, losing).
>
>
> [1] Rafailov, R., Sharma, A., Mitchell, E., Manning, C. D., Ermon, S., & Finn, C. (2024). Direct preference optimization: Your language model is secretly a reward model. Advances in Neural Information Processing Systems, 36.
>
>
> > clear diagrams to illustrate the workflow of the proposed strategy
>
> > The paper does not explain how the test cases in the algorithm are generated.
>
> We have provided a workflow in  Figure 2, and in Section 3, we have explained each step of our proposed strategy in detail. All the prompts we used are shown in the Appendix B.
> In order to make it clear, we give a formal algorithm description of the CodeDPO construction
> pipeline in Algorithm 1 in the Appendix A.
>
> > The paper employs the PageRank algorithm to rank code based on the principles of PLUM, but there are no other innovations related to correctness presented in the paper.
>
> **This is a misunderstanding of our work.**
>
> The PageRank-inspired algorithm and its underlying principles are novel contributions of our paper, developed independently and without reliance on PLUM.
> Importantly, PLUM does not employ a ranking mechanism, which is a core innovation in our approach. Additionally, our results consistently outperform PLUM across multiple benchmarks, further validating the advantages of our method.
>
> We would like to clarify the following points regarding our innovations:
>
> 1. Fundamental Differences in Design Principles:
> Our method introduces a distinct approach to preference learning by defining code preference based on two key factors—**Correctness and Efficiency**. While PLUM focuses on filtering for correctness, we construct an optimization dataset that jointly incorporates these two dimensions. Specifically, for correctness, we propose a novel hypothesis: **Tests executable by more code snippets are more reliable, and code that passes more tests is more likely to be correct.**
> Based on this hypothesis, we designed a self-validation mechanism using a PageRank-inspired algorithm to iteratively score both code solutions and test cases. This approach fundamentally differentiates our work from PLUM and represents a key innovation in our methodology.
> 2. Experimental Validation:
> Our experiments demonstrate the effectiveness of our approach, achieving significant improvements over PLUM across multiple benchmarks. These results further validate the novelty and impact of our proposed algorithm and design principles.
>
>
> > the choice of 15 code solutions and test cases is mentioned, but no justification is provided for this number. Should there not be comparative experiments? Furthermore, why was a temperature of 1.5 chosen for the model?
>
> The choice of sample_num=15 and temperature=1.5 was motivated by practical considerations to balance diversity in the sampled code solutions and test cases. These values were selected based on empirical observations and insights from prior work on data generation.
>
> To address your concern about comparative experiments, we conducted a set of ablation studies to evaluate the impact of different values for sample_num. Specifically, we tested values of 5, 15, and 50, and the experiments follow the design in Section 5.3.1.
>
> We first show the Spearman Correlation between self-validation score and actual code accuracy on HumanEval.
>
> | n    | Spearman |
> | ---- | -------- |
> | 5    | 0.7425   |
> | 15   | 0.8598   |
> | 50   | 0.8613   |
>
> We then present the performance of Phi-2-2.7B with different sample numbers. Similar experimental trends are observed across other models.
>
> |      | HumanEval | HumanEval+ |
> | ---- | --------- | ---------- |
> | n=5  | 54.88     | 49.39      |
> | n=15 | 57.32     | 51.83      |
> | n=50 | 57.90     | 51.83      |
>
> These experiments indicate that sample_num=15 provides a good trade-off between diversity and computational feasibility.

---

> ### Author Response · Authors · 2024-11-21
> **Rebuttal 2**
>
> > economic costs
>
> All the training and inference are served by 16 A100 GPUs. The total cost of our dataset construction process is nearly 80$, with around 40 hours on a server with 32 CPUs, as shown in Appendix B.
>
> > The concept of code efficiency is quite broad. In this paper, efficiency is evaluated solely based on the execution time of test cases, which is clearly insufficient. In practical applications, code efficiency is influenced by various factors such as the production environment and tools used.
>
> The evaluation of code efficiency based on execution time is a fundamental and commonly used metric in both research ([2], [3]) and practical contexts, such as programming competitions and real-world software development scenarios. By following this mature and broadly accepted definition, we ensure that our evaluation aligns with established methodologies in the field.
>
> [2] Shypula, Alexander, et al. "Learning performance-improving code edits." ICLR 2024.
>
> [3] Chen, Binghong, et al. "Learning to improve code efficiency." arXiv preprint arXiv:2208.05297 (2022).
>
> > In Chapter 3, the paper should provide a detailed description of the overall process for training the code, rather than only explaining the process after SFT.
>
> We will add the formula of the training loss for our CodeDPO in Section 3. The training loss of our CodeDPO is defined as:
>
> $L = L_{\text{DPO}} + {L}_{\text{SFT}}$, where the definitions of the DPO loss and SFT loss were presented earlier in the rebuttal.
>
> CodeDPO is designed to be a plug-and-play framework that can be applied to nearly all code models, regardless of their type or pretraining stage. In our experiments, we apply CodeDPO after models have been trained as either base models or SFT models.
>
> **The processes described in Section 3 focus on what happens after the base or SFT model stage, as this is where CodeDPO is applied.** If the concern refers to steps prior to SFT, **we want to note that those steps are not within the scope of CodeDPO itself** but are part of standard pretraining or supervised fine-tuning pipelines.
>
> > The pass rates for CODEDPO and PLUM on the more complex benchmarks MBPP and MBPP+ are similar.
>
> We selected a diverse set of datasets, including HumanEval+ and MBPP+, which are widely recognized and challenging benchmarks. The performance improvements achieved on these datasets are significant and highlight the robustness of our method.
>
> While the pass rates for CodeDPO and PLUM appear similar on some complex benchmarks, it is important to note the following:
>
> **Broader and Consistent Performance**:
>
> CodeDPO consistently outperforms PLUM and other baseline methods across a wide range of benchmarks, including the latest preference optimization (PO) methods like PLUM. We emphasize that PLUM is concurrent work, proposed during a similar timeframe, and not a prior reference.
>
> **Dual Optimization Advantage**:
>
> Unlike PLUM, which focuses solely on correctness, CodeDPO optimizes both correctness and execution time. This dual-objective optimization adds a practical dimension to the generated code, making it more efficient and applicable to real-world scenarios.
>
> **Efficiency and Robustness**:
>
> Our self-generation-and-validation mechanism allows us to generate both code solutions and test cases without relying on strong external models like GPT-4. This approach is not only more robust but also cost-effective, addressing a key limitation of methods like PLUM that depend on computationally expensive resources.
>
> We believe these additional advantages, alongside the consistent improvements demonstrated by CodeDPO, establish its value and novelty compared to existing baselines.
>
> > The paper does not report statistics on the average execution time of the test cases. Is the execution time in milliseconds or seconds?
>
> Below, we provide the average execution time (in seconds) based on experiments conducted with Phi-2-2.7B. The execution times for other models are within a similar range.
>
> Please note that execution time can vary due to differences in computational resources and runtime conditions. To ensure reliability, we conducted multiple experiments in a stable environment and report the averaged statistics as follows:
>
> | Benchmark  | Before CodeDPO (seconds) | After CodeDPO (seconds) | Average Speedup |
> | ---------- | ------------------------ | ----------------------- | --------------- |
> | HumanEval+ | 0.250                    | 0.172                   | 1.45x           |
> | MBPP+      | 0.189                    | 0.137                   | 1.38x           |
>
> These results highlight the consistent improvement in execution efficiency brought by CodeDPO. We will include the detailed statistics in the revised paper for clarity.

---

> ### Author Response · Authors · 2024-11-21
> **Rebuttal 3**
>
> > The experimental strategy in section 5.3.1 lacks theoretical justification and is difficult to understand.
>
> The experimental strategy in Section 5.3.1 explores different preference data construction strategies, designed based on commonly used approaches in related work. These strategies, in addition to our proposed self-validation PageRank algorithm, include the following:
>
> **Filter with All Tests:**
>
> This strategy uses all generated test cases as filtering criteria. Samples that pass all tests are treated as positive examples, while the rest are negative. This approach has been adopted by prior code preference optimization methods such as PLUM, which leverages strong models like GPT-4 for test case generation and filtering. However, this method has limitations:
>
> - It relies heavily on the quality of the generated test cases, which can vary significantly.
> - It often requires the use of powerful models like GPT-4 to generate reliable test cases, increasing computational costs.
> - It may suffer from class imbalance, as the number of positive and negative samples can be skewed.
>
> **Sort by Number of Passed Tests:**
>
> This strategy counts the number of passed tests for each code snippet and uses the samples with the highest and lowest counts as comparison pairs. This principle is commonly employed in post-processing and reranking methods, such as CodeT. However, it also has drawbacks:
>
> - It assigns equal weight to all test cases, failing to account for differences in test case quality.
> - It does not prioritize higher-quality test cases, which limits its ability to emphasize their impact.
>
> By including these strategies, the experimental design in Section 5.3.1 aligns with methods widely used in related work. Through consistency experiments on HumanEval and final performance evaluations, we demonstrate that our self-validation PageRank algorithm outperforms these alternative approaches, offering significant advantages by accounting for test case quality and robustness.
>
> > In section 5.3.2, what strategy was used to construct CODEKTO? Why was CODEKTO chosen for comparison, and why not compare against other PO strategies?
>
> The data construction strategy for CodeKTO follows the same process as for the original CodeDPO, ensuring a consistent and fair comparison.
>
> We chose KTO as a comparison because it is one of the most popular and widely used variants of DPO, particularly in scenarios where pairwise preference data is unavailable. KTO is based on prospect theory and offers a distinct advantage in its ability to work without explicit preference pairs, making it a relevant and meaningful baseline for comparison.
>
> Our decision to include KTO was informed by its prominence in related work and its practical relevance as an alternative preference optimization strategy. While we acknowledge that there are other PO strategies, we believe that including KTO provides valuable insights into how CodeDPO compares against a diverse set of other methods.

---

> ### Author Response · Authors · 2024-11-25
> **Looking forward to your feedback**
>
> Hello, do our responses address your questions? Please let us know if there are any other questions you'd like to discuss!

---

> > ### Author Response · Authors · 2024-12-02
> > **Looking forward to your feedback**
> >
> > Hello, we have revised our paper and add the missing part based on your comment.
> > Do our responses address your questions? Please let us know if there are any other questions you'd like to discuss!

---

### Author Response · Authors · 2024-11-21
**Reply to all Reviewers**

We thank all reviewers for their time and valuable comments.

In this work, we introduce CodeDPO, a framework that integrates preference learning into code generation to improve code correctness and efficiency. CodeDPO employs a novel self-generation-and-validation mechanism along with a PageRank-inspired algorithm to create a code preference optimization dataset.

We would like to thank the reviewers for acknowledging our work to be:

- "The proposed method alleviates the problem of wrong tests when using LLMs generated tests, and the data generation does not require a strong teacher LLM. The page-rank-like algorithm is interesting..." (Reviewer FkPM)

- "The work presents an interesting investigation of DPO in the domain of code generation... this work involves a couple of new ideas, including the PageRank-inspired algorithm and the integration of code efficiency into the optimization objective." (Reviewer bDyW)

- "The paper trains a code generation model with DPO with several pre-trained checkpoints, illustrating the effectiveness of functional-correctness-driven DPO for code generation. The paper proposes a handful of research questions to evaluate CodeDPO on varied benchmarks and compare with multiple baselines." (Reviewer wR3U)

- "The paper focuses on an important topic, LLM-based code generation." (Reviewer xnpY)

Furthermore, we appreciate the reviewers pointing out areas for improvement, and we have prepared substantial additional content. For some reviewers' misunderstandings, we hope to clarify these issues in the detailed responses below.

In the following, we address specific questions for each reviewer.

---

### Meta-Review · Area_Chair_MApE · 2024-12-18

**Metareview:**

The paper proposes CodeDPO, a framework that uses preference learning to improve code generation models in terms of correctness and efficiency. This is an important problem to tackle but, unfortunately, currently the paper is below par for ICLR because of following reasons:

- The proposed approach itself is a straightforward combination of existing methods like DPO. This is not a problem as long as new challenges are mentioned and addressed properly for the specific setting but that doesn't seem to be the case here.

- The experimental evaluation is lacking in multiple dimensions and requires a lot of work. For example, inconsistent dataset sizes between CodeDPO and baselines make comparisons potentially unfair. Even then, the performance improvements are relatively minor or none over existing baselines.

- There were lots of missing details and lack of justification about the approach. For example, specific implementation choices and hyper-parameters were not justified.

Overall, I request the authors to consider improving the paper based on comments from the reviewers for the next cycle.

**Additional Comments On Reviewer Discussion:**

Most reviewers had concerns about unfair comparisons and poor experimental evaluation in the paper. The authors response to these concerns were not convincing. For example, one reviewer remarked about low performance with a baseline which uses less data compared to the proposed approach.
Another reviewer pointed to clear reliance on external resources due to selection of initial seeds from real world repositories. Most reviewers also had concerns about the novelty (I personally don't take this as a big reason to reject).

---

### Decision · Program_Chairs · 2025-01-22

Reject